# Somatic mosaicism and common genetic variation contribute to the risk of very-early-onset inflammatory bowel disease

Eva Gonçalves Serra et al.[#]

Very-early-onset inflammatory bowel disease (VEO-IBD) is a heterogeneous phenotype associated with a spectrum of rare Mendelian disorders. Here, we perform whole-exome-sequencing and genome-wide genotyping in 145 patients (median age-at-diagnosis of 3.5 years), in whom no Mendelian disorders were clinically suspected. In five patients we detect a primary immunodeficiency or enteropathy, with clinical consequences (*XIAP, CYBA, SH2D1A, PCSK1*). We also present a case study of a VEO-IBD patient with a mosaic de novo, pathogenic allele in *CYBB*. The mutation is present in ~70% of phagocytes and sufficient to result in defective bacterial handling but not life-threatening infections. Finally, we show that VEO-IBD patients have, on average, higher IBD polygenic risk scores than population controls (99 patients and 18,780 controls; $P < 4 \times 10^{-10}$), and replicate this finding in an independent cohort of VEO-IBD cases and controls (117 patients and 2,603 controls; $P < 5 \times 10^{-10}$). This discovery indicates that a polygenic component operates in VEO-IBD pathogenesis.

[#]A full list of authors and their affiliations appears at the end of the paper.

nflammatory bowel disease (IBD) represents a heterogeneous group of disorders characterized by a dysregulated immune response toward commensal gut bacteria leading to chronic relapsing intestinal inflammation[1,2]. Crohn's disease (CD) and ulcerative colitis (UC), the two common forms of IBD, affect around 0.5% of individuals of European descent with lower, but rising, prevalence in other parts of the world[3].

Though IBD can occur at any age, the peak age of onset for CD and UC is between 20–30 years and 30–40 years, respectively[4]. About 20% of patients develop IBD before the age of 17 years (pediatric-onset-IBD) and around 14/100,000 children have an IBD onset under the age of six (very-early-onset IBD)[5]. Onset of IBD before the age of 2 years (infantile-IBD) or even within the first month of life (neonatal-IBD) is exceptionally rare. Patients with VEO-IBD present with a higher rate of pancolitis, and subgroups present with severely fistulising disease, resistance to conventional immunosuppressive treatments and immune defects associated with increased lethality[6,7].

Several Mendelian disorders present with IBD and IBD-like intestinal inflammation and have an onset during infancy or within the first 6 years of life[8,9]. Biallelic loss-of-function variants in the *IL10* signaling pathway (*IL10*, *IL10RA*, and *IL10RB*) are fully penetrant for VEO-IBD[10,11], while genetic variants underlying several other Mendelian disorders show an incomplete penetrance of the IBD phenotype (for simplicity called monogenic IBD). The majority of these conditions are autosomal recessive or X-linked inherited primary immunodeficiencies (PIDs). A simple differential diagnosis based on phenotypic associations is often difficult and a sequential work-up based on candidate genes is time consuming. Whole-exome-sequencing (WES) is increasingly used to screen for causal mutations and such studies have identified pathogenic variants in a proportion of VEO-IBD patients, in genes such as *IL10*, *IL10RA*, *IL10RB*, *XIAP*, *TTC7A*, and *TTC37*[12–18]. Identifying the underlying disease-causing variants in VEO-IBD patients is important because it can directly influence patient management and inform on the appropriate treatment strategy, e.g., early haematopoietic stem cell transplantation (HSCT) in patients with haematopoietic defects caused by *IL10*[10], *FOXP3*[19], or *XIAP*[20] mutations.

Rare monogenic forms of IBD are a stark contrast to the polygenic nature of pediatric and adult-onset IBD. Genome-wide association studies (GWAS) have identified more than 240 loci associated with IBD, the majority of which are driven by common variants (minor allele frequency (MAF) > 5%) of small effect (increasing risk by 1.1–1.3 fold), together explaining 13 and 8% of the variance in disease liability for CD and UC, respectively[21–23]. The majority of patients included in these GWAS have adult-onset disease, and very few (<0.1%) have VEO-IBD. Genome-wide association studies focusing on pediatric-onset cases[24,25] identified risk loci that, at the time, were not associated with adult-onset disease but all have since been robustly associated in adult-IBD cohorts[26]. More recently, three studies focusing on patients with an age at disease onset greater than 6 years, reported a weak, but statistically significant, negative relationship between polygenic risk score and the age at CD and UC diagnoses[27–29]. The role of polygenicity in VEO-IBD remains unknown and it is possible that, while rare monogenic variants underlie disease in a proportion of VEO-IBD patients, an exceptionally high burden of common IBD-susceptibility alleles may also contribute to VEO-IBD risk. Such a hypothesis has not been previously investigated in VEO-IBD cohorts.

Here, we use WES and genome-wide SNP arrays to better understand the genetic architecture of VEO-IBD in a multi-center cohort of 145 patients with a median age-at-diagnosis of 3.5 years and a severe disease course, indicated by previous surgery or need for biological therapy. We include patients in whom a Mendelian disorder is unexpected due to clinical presentation, or were mutation negative following screening of specific VEO-IBD genes selected based on patient presentation (e.g., *IL10*, *IL10RB*, or *IL10RA* defects in patients with IBD onset in the first three months of life). The cohort is therefore potentially enriched for cases harboring undiscovered monogenic causes of VEO-IBD, or alternative causal mechanisms. We investigate the extent to which mutations in 67 known monogenic IBD genes account for disease in this selected cohort and search for novel monogenic causes exome-wide. Moving beyond rare variation, we use genome-wide SNP data to evaluate the role of common CD and UC-susceptibility alleles in the pathogenesis of VEO-IBD. By generating polygenic risk scores (PRS) based on the effect-size estimates of SNPs significantly associated with adult-onset CD and UC and replication in independent VEO-IBD and control cohorts, we investigate whether VEO-IBD children harbor a higher load of such alleles when compared to a large collection of adult-onset IBD cases or population controls (see Supplementary Fig. 1 for an overview of the study workflow).

## Results

**Cohort sequencing and quality control.** A cohort of 145 VEO-IBD individuals and 4436 population controls were exome sequenced at a mean coverage of 69× and 53×, respectively. Following sample and variant-based quality control (see Methods section), 145 VEO-IBD cases and 3969 controls with equivalent sequencing-based QC metrics (Supplementary Fig. 2) remained for analysis, with an average of ~40,000 variants called per exome and 94% of genes covered at a mean depth of 30× or above (Supplementary Fig. 3). There were two well defined ancestry-matched groups within our dataset: 104 cases and 3787 controls defined as being of European descent and 21 cases and 68 controls defined as South Asian (the remaining samples did not cluster with a clearly defined ancestry group) (see Methods section, Supplementary Fig. 4).

**Somatic mosaicism: non-Mendelian inheritance of VEO-IBD.** The initial screening for pathogenic variants in established monogenic IBD genes identified a nonsense mutation in *CYBB* (p.W380X) in a 31-year-old male patient of European descent with infantile-onset of granulomatous colitis, perianal abscesses and hidradenitis suppurativa (Fig. 1a, f; Table 1). A detailed clinical summary of the case is presented in Supplementary Note 1 and Supplementary Fig. 5. Loss-of-function mutations in *CYBB* are known to cause chronic granulomatous disease (CGD). However, our patient had no history of invasive infections, a potential fatal hallmark of CGD unless patients are closely monitored and rapidly treated. We found no carriers of the mutation in more than ~156,000 whole-exome sequences from ethnically-diverse population controls (including ExAC[30] and gnomAD[30]).

*CYBB* resides on the X-chromosome and thus we were surprised to notice that only 122 out of 174 sequence reads (~70%) covering the site of the mutation carried the nonsense allele (A) (the other reads carried the wild-type allele—G). Investigating the common genetic variation across the X chromosome, we found no evidence of Klinefelter Syndrome (47 chromosomes, XXY) or partial X-chromosomal duplication that could explain this observation. We confirmed the mutation via Sanger sequencing undertaken in an independent clinical genetics laboratory. We also Sanger-sequenced DNA from the patient's mother and sister and established that neither are carriers of the nonsense mutation (Fig. 1b), suggesting that the mutation is a de novo event of hemizygous mosaicism. Accordingly, 70% of neutrophils lacked CYBB protein expression, further supporting a hemizygous mosaicism (Fig. 1c). Functional

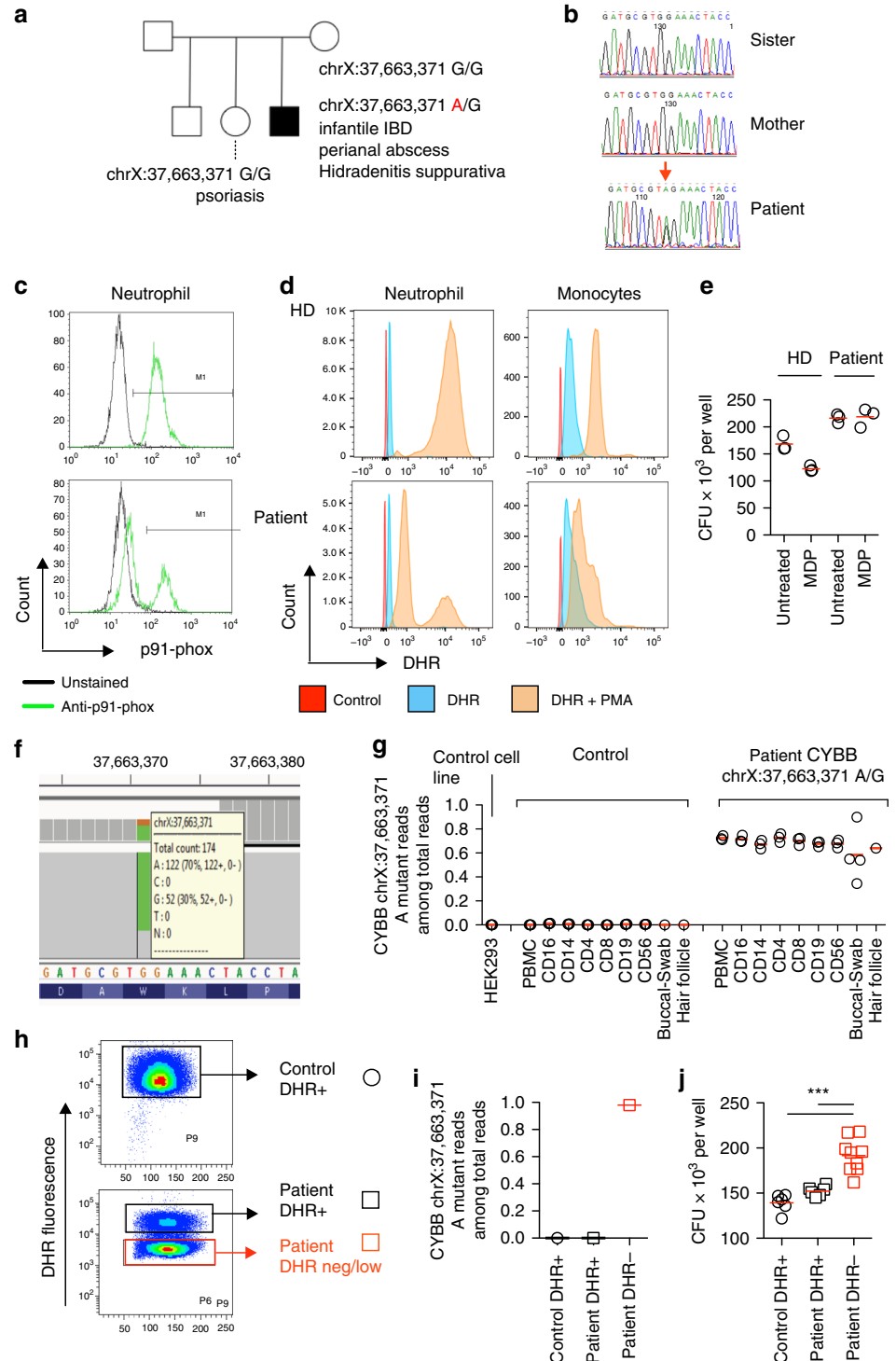

validation experiments confirmed the mosaicism, with 70% of neutrophils and monocytes showing completely absent NADPH-oxidase activity as seen in CGD patients, and 30% of cells showing a normal profile (Fig. 1d, see Methods section). This degree of mosaicism affected bacterial handling capacity since monocyte-derived macrophages from the patient had ~30% more colonies at baseline but did not respond to muramyl dipeptide (MDP) in the gentamicin protection assay (Fig. 1e; see Methods section), a finding suggestive of a bacterial handling defect[31]. Our results suggest that the proportion of cells with complete loss-of-function of CYBB affects the phenotypic presentation, i.e., 30% wild-type

neutrophils is sufficient to prevent life-threatening infections but 30% fully-functioning macrophages is insufficient to prevent intestinal inflammation as a result of ineffective bacterial killing and inflammatory cytokine production.

To further understand the origin and the developmental time at which the mutation arose during embryogenesis, we sequenced DNA from several flow cytometry assay (FACS) sorted immune cell subsets and used DNA from buccal swabs, containing epithelial cells, and hair follicles from the patient and compared those with wild type cells (Fig. 1f, g; see Methods section, Supplementary Fig. 6). In all immune cell subsets of the patient

**Fig. 1 Analysis of *CYBB* mosaicism in a male patient. a** Pedigree structure for the family of the male patient with the mosaic hemizygous mutation in *CYBB* (chrX:37,663,371A/G; p.W380X). **b** Sanger sequencing of the chrX:37,663,371 *CYBB* mutation site in the patient and unaffected relatives (sister and mother). **c** p91-phox protein expression (the gene product encoded by *CYBB*) analysed by flow cytometry assay (FACS). Control is a healthy donor. **d** Measurement of oxidative burst in neutrophils and monocytes using the dihydrorhodamine-1,2,3 (DHR) assay. Obtained from the patient and a healthy donor (control). **e** Defective bacterial handling in monocyte derived macrophages with the *CYBB* mosaicism. Intracellular survival of *Salmonella typhimurium* was quantified using the agar plate technique. Results show three technical replicates. Obtained from the patient and a healthy donor (control). **f, g** Quantification of mutant read proportion at chrX:37,663,371 using the IGV browser. PBMCs were sorted into immune cell subsets (Supplementary Figs. 6B, C) and compared with buccal swabs and hair follicles, as well as with healthy donor immune cells and a HEK293T cell line as technical control. **h** FACS sorting strategy for DHR-high and DHR-low populations following DHR staining and PMA stimulation (Supplementary Fig. 6A). **i** Quantification of mutant reads at chrX:37,663,371 following sorting based on DHR for control DHR-high, patient DHR-high, and patient DHR-low neutrophils (Supplementary Fig. 6A). **j** Gentamicin protection assay on neutrophils for control DHR-high, patient DHR-high, and patient DHR-low populations (Supplementary Fig. 6A). Briefly, neutrophils were infected at a MOI 1:10 for 45 min with *Salmonella enterica serovar typhimurium* and subsequently treated with gentamicin for 45 min. Neutrophils were then lysed and plated on LB agar plates for CFU counting on the following day. ***$p < 0.001$, Mann–Whitney *U*-test.

(neutrophils, monocytes, CD4+ T cells, CD19+ B cells, NK cells) we found the mutation was present in around 70% of cells (range: 63–76%), whereas in buccal swabs it was carried by around 60% (range: 34–90%) of cells. In order to ensure that the approximately 70% of phagocytes that do not produce reactive oxygen species correspond to the mutated ones, we sorted DHR-high and DHR-low neutrophils from the patient (Fig. 1h, Supplementary Fig. 6). As expected, the DHR-high cells of the patient were 100% wild type, whereas 98% of the DHR-low cells carried the pathogenic allele (Fig. 1i). In line with the mutated genotype, the DHR-low population corresponded to a higher number of bacterial colonies following *Salmonella* Typhimurium infection compared to both DHR-high cells from the patient and control (Fig. 1j), suggesting a clear genotype association with antimicrobial capacity. Our results indicate that both mesodermal (haematopoietic cells) and ectodermal (cheek epithelia and hair follicles) derived cells are affected and that the mutation arose early in embryogenesis, likely between days two and five, and certainly before day nine, the time at which the mesoderm and ectoderm separate. This case study provides evidence that nonclassical modes of inheritance, such as somatic mosaicism, can underlie VEO-IBD.

**Primary immunodeficiencies as a cause of VEO-IBD.** We identified four patients with an underlying primary immunodeficiency. Among the genes screened (Supplementary Table 2) there were two hemizygous nonsense alleles in *XIAP* (p.R222X) and *SH2D1A* (p.R75X) in two patients (Table 1) of European and African descent, respectively, and a homozygous missense change in *CYBA* (p.S118N), shared by two South Asian siblings (Table 1). These variants change well conserved amino acids (GERP scores = 3.8, 2.7, 4.5, respectively), have high CADD scores (37, 38, 25, respectively), and there are no recessive carriers of these alleles in more than 156,000 ethnically-diverse population sequences (including ExAC and gnomAD), indicating that these mutational events are exceptionally rare in the population (Table 1). A list of likely benign variants, or variants of uncertain significance identified in the 67 screened monogenic IBD genes can be found in the Supplementary Note 2. The majority of the 67 genes were sequenced at a depth (mean = 67×) comparable to the rest of the exome (mean = 69×, Supplementary Fig. 3). In keeping with previous reports[13], two of the genes (*NCF1* and *IKBKG*) had poor coverage (<10×) following WES, and thus pathogenic variants in these genes may have been missed.

Mutations in *XIAP* have been associated with X-linked lymphoproliferative syndrome 2 (XLP2, MIM: 300635). The premature stop mutation identified in *XIAP* (p.R222X) disrupts the region responsible for the XIAP-RIPK2-NOD2 interaction in the BIR2 domain as well as the C-terminal part of the enzyme required for ubiquitin ligase activity. Loss-of-function was experimentally confirmed by absent TNF response after muramyl dipeptide stimulation (Fig. 2a).

*SH2D1A* mutations have been causally implicated in X-linked lymphoproliferative syndrome 1 (XLP1, MIM: 308240). The *SH2D1A* nonsense allele (p.R75X) detected in our patient truncates the protein in the middle of the SH2 domain, the critical region for signal transduction[32]. The truncation in *SH2D1A* was confirmed by FACS, showing absent binding of a C-terminal detection antibody (Fig. 2b). The patient died due to severe EBV infection and liver failure, a severe phenotype previously described in XLP1 patients[33].

Mutations in *CYBA*, which encodes p22phox, cause chronic granulomatous disease (CGD, MIM: 233690). The p.S118N variant observed in both our patients is located within the putative membrane-spanning domain of the protein, where the majority of missense pathogenic variants have been found[34–36]. Functional impairment was confirmed using the dihydrorhodamine-1,2,3 (DHR) assay using several stimuli (formylpeptide, E. coli and PMA; Fig. 2c), which showed impaired superoxide production in neutrophils from both siblings. A similar homozygous mutation in *CYBA* (p.S118R) has been described previously in a patient with CGD[37].

**Exome-wide screening for recessive loss-of-function variants.** In addition to screening for genetic defects that have previously been described in patients with IBD-like intestinal inflammation, we searched exome-wide, for homozygous, or potential compound heterozygous, or hemizygous essential loss-of-function variants in our VEO-IBD cohort. This analysis revealed a homozygous nonsense variant in *PCSK1* (p.R391X), which was absent from gnomAD and affected a highly conserved nucleotide (GERP score of 5.4). *PCSK1* (Proprotein Convertase Subtilisin/ Kexin type 1) encodes the proprotein convertase enzyme which cleaves prohormones and defects in the gene have been linked to an endocrinopathy syndrome characterized by diarrhea but rarely intestinal inflammation. The patient harboring the *PCSK1* variant was of Asian ancestry and presented with indeterminate mild colitis before the age of one when recruited to the study. Interestingly, the initial intestinal inflammation did not progress but the phenotype changed over time. After recruitment and submission of the DNA sample for sequencing, the phenotype evolved towards growth delay, excessive weight gain, and endocrine disorders including diabetes and hypothyroidism, hypogonadism, cryptorchism, cortisol deficiency, and chronic lung disease. Whereas the initial phenotype was uncharacteristic, the subsequent syndromic findings are fully explained by *PCSK1* deficiency. This finding highlights the value of next generation sequencing as a predictive diagnostic tool, as well as the need to take phenotype progression into account. No other likely essential loss-of-function variants were identified in our cohort.

**Table 1 Pathogenic variants identified in VEO-IBD patients upon screening of known IBD-associated Mendelian disorder genes.**

| Gene | Position | Variant | GTs | GERP | CADD | ADD | IT | GM | Ethnicity | Phenotype |
|---|---|---|---|---|---|---|---|---|---|---|
| XIAP | X:123020176 | ENST00000371199.7: c.963C>T ENSP00000360242:p.R222* | Hem | 3.8 | 37 | 6 | 0 | 0 | EU | CD-like phenotype with a severe fistulizing perianal phenotype |
| SH2D1A | X:123504047 | ENST00000371139.8: c.522A>T ENSP00000360181:p.R75* | Hem | 2.7 | 38 | 6 | 0 | 0 | African | Acute EBV infection and liver failure |
| CYBA | 16:88712540 | ENST00000261623.8: c.492G>A ENSP00000261623:p.S118N | Hom | 4.5 | 25 | 5, 5* | 0 | 1 | South Asian | Granulomas and a non-stricturing, non-penetrating CD-like pathology |
| CYBB | X:37663371 | ENST00000378588.4: c.1206G>A ENSP00000367851:p.W380* | Hem | 5.6 | 40 | 0 | 0 | 1 | EU | CD (perianal disease), Hidradenitis suppurativa |

Each row represents a variant in a conserved site (GERP > 2), predicted damaging by in-silico tools, identified in VEO-IBD cases. Patient genotypes (GTs) are listed (Hem: hemizygous, if male; Hom: homozygous for the alternative allele). The number of gnomAD (GM) and INTERVAL (IT) samples that harbored similar variants in that gene (i.e., nonsense alleles) with the same genotype as our patients are also listed. Patient ADD refers to age at diagnosis (in years). Ethnic origin of patients as confirmed via PCA analysis (EU European descent). CADD scores in table refer to C-scaled scores. CD Crohn's disease. All variants were functionally validated. All variants were absent from gnomAD and INTERVAL datasets, and therefore constituted novel variants herein identified
*The variant in CYBA was identified in two siblings. Genomic positions based on GRCh37.

**Searching for VEO-IBD genes: gene-based analysis.** To identify previously unreported genes involved in VEO-IBD, we searched exome-wide for genes with a significant difference in the burden of rare variants in our cases versus a large cohort of exome-sequenced controls ($N = 3855$ INTERVAL samples; see Methods section). This approach has the advantage of allowing variants across the penetrance spectrum to contribute to the association test (see Methods section). Nine different exome-wide, gene-based screens were conducted using different variant inclusion criteria (for variant severity and minor allele frequency; see Methods for definitions and the Methods). No individual gene achieved exome-wide significance (Fisher's exact test $P < 1.7 \times 10^{-6}$ after correction for multiple testing) irrespective of the variant inclusion criteria (Supplementary Fig. 7).

**Testing for a rare variant burden in pathways and genesets.** We next searched for a rare variant enrichment across multiple related genes, such as those that reside in the same biological pathway (see Methods section and Supplementary Table 3). This approach offers additional statistical power compared to individual gene tests, as a larger number of variants are collapsed across a larger testing unit. In total we tested 195 different biological genesets, 186 of which represent the whole set of KEGG pathways available in the KEGG pathway database. No geneset or pathway showed a significant burden of rare variants in patients versus controls ($N = 3855$ INTERVAL samples) after correction for multiple testing (PLINK/SEQ burden test statistic $P < 3.2 \times 10^{-5}$, Supplementary Fig. 8).

**No evidence of a rare variant burden in PID loci in VEO-IBD.** A previous WES study of 125 VEO-IBD children and 145 healthy controls[38] reported an over-representation of rare, damaging variation (AF < 0.1%) in PID-associated genes in VEO-IBD patients ($P < 1 \times 10^{-4}$). We found no evidence (PLINK/SEQ burden test statistic $P = 0.7$) of such an enrichment in our VEO-IBD cohort when compared against a larger set of control sequences ($N = 3855$ INTERVAL samples; Supplementary Fig. 8) and while controlling for potential confounding factors (such as ancestry and sequencing depth) between cases and controls (see Methods section).

**A polygenic component operates in VEO-IBD risk.** We next searched for a polygenic component underlying the disease using our genome-wide genotyping data. We calculated a PRS for each VEO-IBD COLORS patient of European ancestry ($N = 99$) by weighting their risk allele count at each disease-associated SNP ($N_{CD} = 147$; $N_{UC} = 119$) by the estimated effect size of the risk allele (see Methods section). The CD and UC risk polygenic risk scores were significantly greater in our VEO-IBD cases compared to a large cohort of population controls ($N = 18,780$) (Student's t-test $P = 3.97 \times 10^{-10}$ and $P = 1.23 \times 10^{-10}$ for CD and UC, respectively) (Fig. 3). We did not detect a significant difference between the risk scores of our VEO-IBD cases and a cohort of 13,896 predominantly adult-onset IBD cases ascertained by the UK IBD Genetics Consortium (Student's t-test $P = 0.457$ and $P = 0.661$ for CD and UC, respectively) (Fig. 3). We can conclude that if there is a difference in mean PRS between VEO-IBD and adult-onset disease then it must be smaller than that seen between adult-onset IBD cases and controls (Cohen's $d = 0.85$ and $0.64$ for CD and UC, respectively) because our sample sizes would provide 100% power to detect such an effect. However, with our sample sizes we cannot rule out the possibility of a smaller difference in mean PRS between VEO-IBD and adult-onset IBD cases (Cohen's $d = 0.2$) because 310 VEO-IBD cases would be required to provide 80% power to detect such an effect, given our sample size of

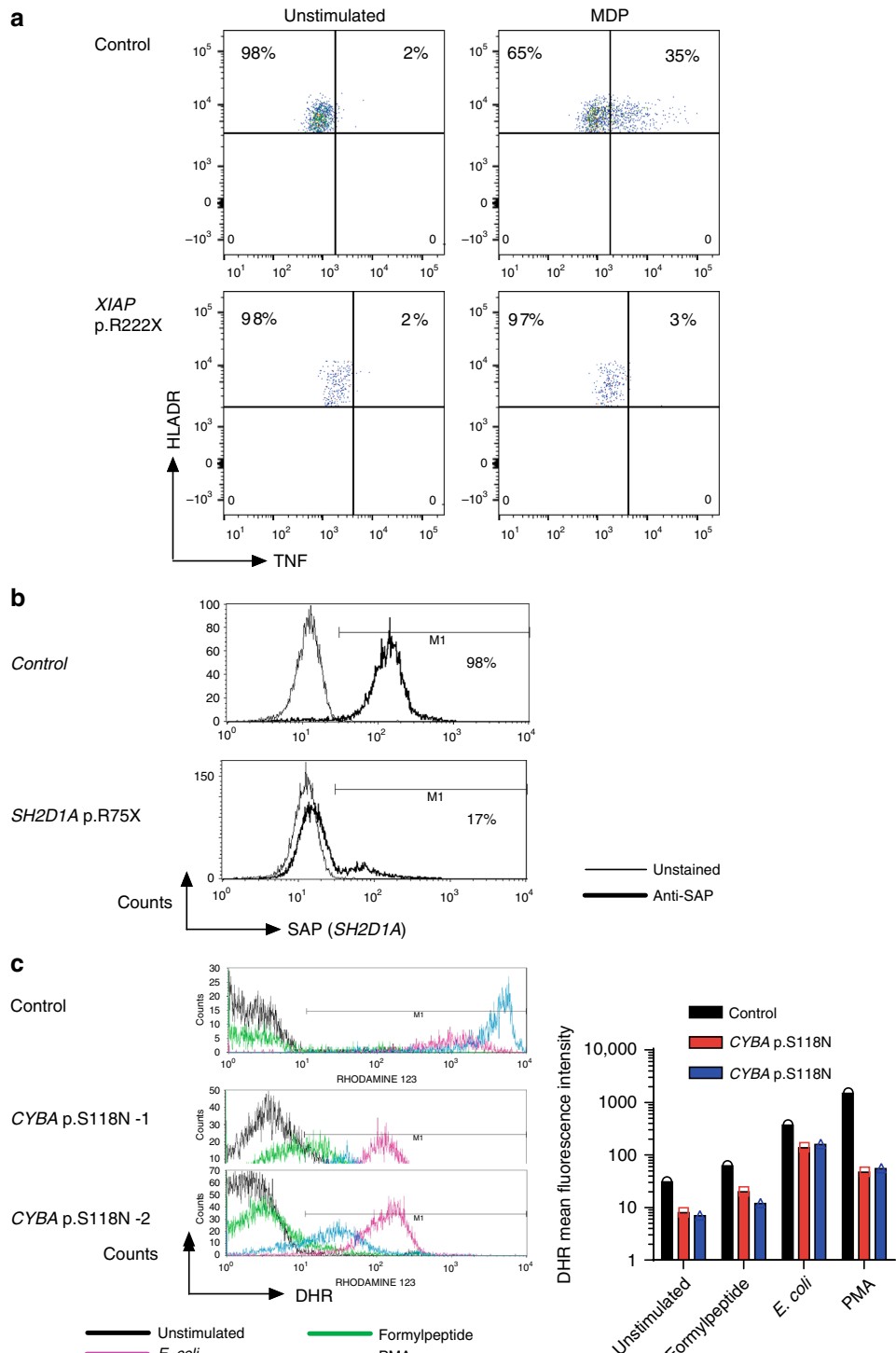

**Fig. 2 Functional validation of pathogenic variants identified in monogenic IBD genes. a** Defective MDP response in a patient with hemizygous *XIAP* p. R222X. MDP (muramyl dipeptide) induced intracellular TNF response was determined using FACS. **b** Absent SAP staining (gene product of *SH2D1A*) as indicated by C-terminal antibody in a patient with hemizygous *SH2D1A* p.R75X. Measured with fluorescence-activated cell sorting (FACS). **c** Defecting ROS production in neutrophils from patients with homozygous *CYBA* p.S118N variants. Dihydrorhodamine-1,2,3 (DHR) flow cytometry assay (FACS) was performed to measure NADPH oxidase activity in response to PMA, E. coli particles and formylpeptide.

adult-onset patients and an α of 0.01 (Supplementary Fig. 9). Our PRS results were not driven by the few loci known to be associated with age-at-diagnosis of IBD (*NOD2*, *HLA*, and *MST1*)[27] and similar findings were obtained when restricting the CD polygenic burden test to VEO-IBD cases defined in the COLORS cohort as CD or CD plus undeterminable IBD (IBDu), and when

restricting the UC polygenic burden test to VEO-IBD cases defined as UC or UC plus IBDu (Supplementary Fig. 10).

To validate these results we generated polygenic risk scores across another cohort of European-descent VEO-IBD cases (Toronto SickKids VEO-IBD, $N = 117$) and population controls (NIDDK, $N = 2603$) with existing genotype data (see Methods

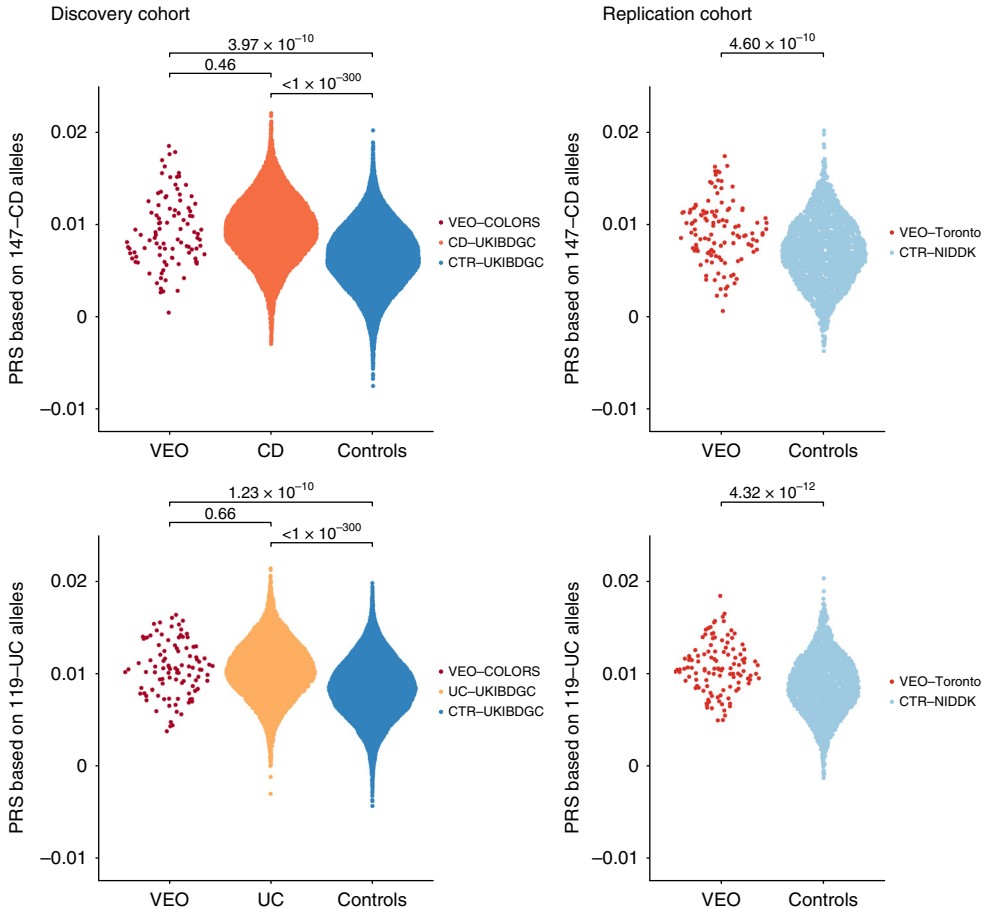

**Fig. 3 Distribution of CD and UC risk scores in VEO-IBD, CD, UC cases and healthy controls.** The CD score was calculated using 147 CD risk alleles and the UC score using 119 UC risk alleles. Both scores were generated for a discovery cohort comprising 99 VEO-IBD cases (VEO-COLORS), 7578 CD cases, 6318 UC cases, 18,780 UK population controls (from the UKIBDGC), all of European ancestry. The replication cohort comprised 117 VEO-IBD cases (VEO Toronto) and 2603 population controls (from the NIDDK Genetics Consortium). The CD and UC risk scores did not significantly differ between the two VEO-IBD cohorts (CD $P = 0.98$; UC $P = 0.64$). The Student's $t$-test was used in group comparisons.

section). We replicated our discovery of a polygenic component operative in VEO-IBD (Student's $t$-test $P = 4.60 \times 10^{-10}$ and $P = 4.32 \times 10^{-12}$ for CD and UC, respectively, Fig. 3).

We were unable to separately quantify the contribution of common IBD risk variants to VEO-IBD in the presence of a Mendelian disease because only one COLORS VEO-IBD individual included in the PRS analysis had a Mendelian diagnosis (the other four patients with Mendelian diagnoses were of non-European descent and were thus excluded from the polygenic analysis). Using a cohort of exclusively VEO-IBD patients, our results demonstrate that a polygenic component operates in VEO-IBD.

## Discussion

Using WES in a selected cohort of VEO-IBD patients in whom no Mendelian disorders were clinically suspected, we identified five patients with rare, pathogenic mutations in four PID-associated genes (*XIAP*, *CYBA*, *CYBB*, and *SH2D1A*), one of which was a de novo mosaic variant (p.W380X in *CYBB*). We assessed the pathogenicity of the variants according to the stringent ACMG and AMP criteria[39], which requires the variant to be absent or infrequently observed in large population reference cohorts, be genetically and phenotypically consistent with previous literature, and have a measurable effect on protein function.

The discovery of these Mendelian variants had a profound impact on the treatment of the surviving four patients. For instance, HSCT was initiated in the patient diagnosed with *XIAP* deficiency. This patient experienced multiple gastrointestinal operations and wound healing problems but has not yet developed EBV-triggered immunopathology (e.g., hemophagocytic lymphohistiocytosis). Routine infection screening to detect this potentially fatal complication that commonly develops in XLP2 patients will now be part of his clinical follow-up. Furthermore, the two siblings with pathogenic variants in the NADPH oxidase genes (*CYBA*) were referred to an immunology clinic and stem cell transplantation is currently being evaluated. The genetic diagnosis of *SH2D1A* deficiency was confirmed by the XLP1-specific extraintestinal complications of EBV-driven disease and liver failure. The importance of early genetic screening in VEO-IBD children to allow assessment before full manifestation of complications, or need for surgery[9,13], was highlighted by the lethal outcome for the patient with *SH2D1A* deficiency (XLP1, p.R75X) while WES was being undertaken in this patient.

The patients with *XIAP*, *SH2D1A*, and *CYBA* defects and the patient with *CYBB* mosaicism illustrate the opportunities and challenges with genetic diagnosis in patients with extreme phenotypes of IBD, where rare genetic variants can cause both immunodeficiency, such as invasive infections, and tissue inflammation. In light of the variable manifestation of the phenotype (some patients with IBD have not presented with the classical immunodeficiency phenotypes) the intestinal inflammation in these patients might either be regarded as a manifestation of the

immunodeficiency (such as XIAP deficiency, XLP1, or chronic granulomatous disease) or as a form of monogenic IBD.

Somatic mosaicism in *CYBB* has previously been reported in two patients with clinical presentation of chronic granulomatous disease, including liver and perianal abscesses and lymphadenitis[40]. Yamada et al. identified the mosaic events following functional tests of neutrophil function warranted by the presentation of CGD. Our study identified the somatic mutation in *CYBB* via whole-exome sequencing and showcases how this approach can be used to find pathogenic mutations that are not indicated by the clinical presentation of a patient. The two patients reported by Yamada et al. only had 0 and 1.6% wild-type PBMCs (but 1.8 and 18.8% wild-type cells from buccal swabs) and were diagnosed with CGD. Our patient had 30% wild-type PBMCs and showed no symptoms of CGD, suggesting that having around ~30% functional phagocytes is sufficient to prevent these life-threatening infections (but >2% is insufficient). This is an important insight into the potential utility of gene therapy for treatment of CGD; correcting the genetic sequence in around 30% of phagocytes could be sufficient to prevent serial life-threatening infections, but unlikely to reduce the risk of intestinal inflammation. The mosaic patient herein reported adds to an expanding spectrum of NADPH oxidase deficiencies where the degree of functional NADPH oxidase defect confers variable penetrance of intestinal and extraintestinal manifestations such as skin inflammation[36,41,42].

It remains to be seen if the extent to which the polygenic background influences VEO-IBD susceptibility varies depending on the monogenic "foreground". For example, the polygenic burden of IBD risk alleles may be of little phenotypic consequence in the presence of fully penetrant VEO-IBD mutations (such as those seen in the IL10 signaling pathway). This model would match that seen in patients with diabetes, where individuals with monogenic MODY have lower polygenic risk score compared to patients with type 1 diabetes[43]. However, for patients with a monogenic VEO-IBD diagnosis that demonstrates incomplete penetrance for VEO-IBD, the polygenic burden of IBD risk could play a more significant role in defining the phenotype. The recent observation that patients with chronic granulomatous disease (CGD) and intestinal inflammation have a greater burden of classical IBD risk variants compared to CGD patients without intestinal inflammation[44] supports this general hypothesis. We show that, at least for individuals with VEO-IBD and no current genetic Mendelian diagnosis, common genetic variants associated with susceptibility to Crohn's disease and ulcerative colitis do play a role in disease. It has recently been shown that there is a weak, but statistically significant, inverse correlation between age of IBD onset and burden of IBD associated risk alleles[27–29]. We found no evidence to support the hypothesis that VEO-IBD is due to an increased burden of common IBD susceptibility alleles relative to adult-onset disease, but larger sample sizes than ours are required to powerfully search for small differences in PRS. This model applies in familial hypercholesterolemia, a dominantly-inherited disorder where individuals with very high LDL-cholesterol and no known monogenic cause of disease have a particularly high burden of common cholesterol-increasing alleles[45]. The recent identification of common genetic variants that are associated with Crohn's disease prognosis but not susceptibility also suggests that the search for disease age-at-onset loci should be performed genome-wide and not just restricted to known IBD susceptibility loci[27].

In summary, our data show that primary immunodeficiencies caused by rare genetic variants can be found in some VEO-IBD patients even if no Mendelian disease was clinically suspected, suggesting that genetic screening is relevant across this entire patient group. We implicate cellular mosaicism with Mendelian

disorder-associated variants as a possible mechanism underlying VEO-IBD, as highlighted by our case study. Finally, we show that whatever factors are driving an early age at disease onset in individuals without a conclusive Mendelian diagnosis, in the majority of patients they do so on a polygenic background similar to classical IBD.

## Methods

**Patient and controls samples**. The study was approved by the North Staffordshire Research Ethics Committee (REC: 09/H1204/30; subproject COLORS in IBD) and local ethics committees at the study sites. All patients, or their parents, gave written informed consent before enrollment. The somatic mosaic patient consented to open access publication of a detailed case report including genetic, clinical, laboratory data as well as endoscopic and histological images. The cohort consists of 146 VEO-IBD cases (singletons) without a previous genetic diagnosis, recruited as part of the COLORS IN IBD project (COLitis of early Onset Rare disorderS in IBD). Samples were referred from participating centers in the United Kingdom, Switzerland, Poland and Germany. All patients had a confirmed diagnosis of IBD by standard methods, including endoscopic, radiologic, laboratory, and clinical evaluation (ESPGHAN guidelines[46]). Phenotypic status was based on the Paris Classification[47]. Patients were selected according to age-at-diagnosis (<7 years, age of symptom onset <6 years) and the severity of the IBD phenotype, as indicated by need for surgery and/or therapy progression to biologics or immunomodulators. When a clinical diagnosis of a known Mendelian disease was suspected (e.g., *IL10*, *IL10RB*, or *IL10RA* defects in patients with IBD onset in the first three months of life), candidate genes were pre-screened by a clinical genetics laboratory. If a genetic diagnosis was established, the individual was excluded from our study in an attempt to enrich the cohort with cases harboring undiscovered monogenic causes of VEO-IBD. Detailed demographic and phenotypic characteristics of the VEO-IBD cohort are provided in Supplementary Table 1. Whole-exome sequence data from 4436 healthy individuals from the INTERVAL Study (www.intervalstudy.org.uk) were ascertained for use as controls in exome-based analyses[48].

**Exome-sequencing of cases and controls**. VEO-IBD cases and INTERVAL controls were sequenced at the Wellcome Trust Sanger Institute (WTSI). Genomic DNA (1–3 μg) was sheared to 100–400 bp using a Covaris E210 or LE220 (Covaris, Woburn, Massachusetts, USA). Sheared DNA was subjected to Illumina paired-end DNA library preparation and enriched for targeted sequencing using the SureSelectXT Human All Exon kit (Agilent Technologies, Santa Clara, CA, USA; Human All Exon 50 Mb—ELID S04380110) according to the manufacturer's recommendations (Agilent Technologies, Santa Clara, CA, USA; SureSelectXT Automated Target Enrichment for Illumina Paired-End Multiplexed Sequencing). Enriched libraries were sequenced (eight samples over two lanes) using the HiSeq 2000 platform (Illumina) with paired-end 75 base reads, according to the manufacturer's protocol. The Burrows-Wheeler Aligner[49] was used for alignment to the human reference genome build UCSC hg19/GRCh37 (1000Genomes_hs37d5). Variants were first called on a per sample basis using GATK Haplotype Caller (version 3.4) and then joint-called across all cases and controls using GATK CombineVCFs and GenotypeVCFs using default settings[50].

**Sample QC**. Cross-sample contamination was evaluated using VerifyBAMID (version 1.1.0). No case samples showed evidence of contamination, however 113 controls had a FREEMIX fraction > 3% and were thus excluded from the study. Samples with a mean genotype quality (GQ) < 85.4 (representing 3 s.d. from the mean), a depth < 40, a missing genotype rate > 0.2%, or controls with a close familial relationship (Pihat > 0.125) were removed from further analysis. A total of 468 poor quality samples (one case and 467 controls) were excluded from subsequent analyses.

**Ancestry analysis**. Cases and controls were assigned to ancestry-matched groups based on principal components inferred from the 2504 individuals in the 1000 Genomes Project (1KG) phase 3 data[51]. Overall, 104 cases and 3787 controls were defined as Europeans, and 21 cases and 68 controls as South Asians. The remaining 20 cases and 115 controls did not cluster with a clearly defined ancestral group. In total, 145 of 146 cases and 3969 of 4436 controls remained after QC (Supplementary Fig. 4).

**Variant QC**. Individual variants were evaluated using Variant Quality Score Recalibrator (VQSR) using the recommended training sets[50] and a 99.9% sensitivity tranche. Individual genotypes with GQ < 20 or depth < 8 were set to missing[52]. Variants with more than 10% missing genotypes in either cases or controls were also removed. In total, 1,267,058 variants passed QC.

**Variant annotation**. The exome data was annotated using several resources: dbSNP v137 rsIDs and allele frequencies (AFs) from 1KG phase 1 ($N = 2818$)[53], NHLBI GO Exome Sequencing Project 6500I (ESP, $N = 6500$)[54], the UK10K low-coverage study (UK10K_WGS, $N = 3781$), UK10K WES samples (UK10K_WES,

$N = 4975$[55], the Exome Aggregation Consortium r0.3 (ExAC, $N = 60,706$)[30] and the Genome Aggregation Database (gnomAD, $N = 141,356$)[30]. Functional annotations were added using the Ensembl Variant Effect Predictor (VEP version 75)[56] according to Gencode v19 coding transcripts, using the most severe consequence on the gene. Deleteriousness scores for missense variants were inferred using the SIFT[57] and PolyPhen-2[58] algorithms. Conserved amino acids were identified using GERP[59].

*Rare variants* were defined as those that were absent or had AFs < 1% in reference datasets. We defined two broad categories of variants, based on predicted functional impact at the protein level: (1) Functional: transcript ablation, stop gained/lost, stop retained, splice donor/acceptor/region, frameshift, inframe insertion/deletion, initiator codon, and missense variants and (2) Disruptive: nonsense, frameshift and splice acceptor/donor variants or missense variants with a PolyPhen-2 and SIFT pathogenicity predictions of "possibly damaging/deleterious" (or greater) and a GERP score > 2.

**Screening of known IBD-associated Mendelian disorder genes**. A list of 67 genes known to be associated with Mendelian disorders with IBD-like inflammatory phenotypes was ascertained from the literature[8,9,60]. The mode of inheritance and associated disorders for these genes are listed in Supplementary Table 2. We screened these genes for rare disruptive variants consistent with the established mode of inheritance. For variants on the X chromosome, we also assessed possible instances of non-classical inheritance, such as non-random X-inactivation events in females, or somatic mosaicism in males. To assess the pathogenicity of the variants identified and to adhere to current best practices[39,61], we checked that the patient's phenotype data matched that reported for the Mendelian disorder, and performed functional validation assays to measure the biological impact of the identified mutations. Genetic variants were classified according to the five-tier terminology system recommended by the American College of Medical Genetics and Genomics (ACMG) and the Association for Molecular Pathology (AMP)[39]. All pathogenic and likely pathogenic variants were validated by an independent clinical genetics laboratory using Sanger Sequencing. We investigated whether any of the identified rare variants were already described as causative for the respective Mendelian disorder based on searches of the literature and the Human Gene Mutation Database (HGMD).

For gene-wide sequencing performance, sequencing coverage per gene was calculated by extracting the read depth at each nucleotide (within exonic sequences of each of the 67 genes) on a per-sample level using BAM files and SAMtools *mpileup*. The read depth was then averaged per-coordinate across samples to produce an average capture per position, as well as a mean and median coverage per gene. The exonic regions of genes were defined by Ensembl v75 (Supplementary Fig. 3).

To identify novel VEO-IBD associated genes, we conducted a gene-based case–control analysis testing for an enrichment of rare variants in VEO-IBD cases versus controls using the Fisher's exact test implemented in PLINK/SEQ (https://atgu.mgh.harvard.edu/plinkseq). We tested four different AF thresholds (<1, <0.5, <0.1% and unique variants) across our two different functional categories (functional and disruptive variants), plus one test for functional <1% homozygous variants only (to avoid diluting a genuine recessive signal). The significance of each gene was empirically assessed by permuting case–control labels 20,000 times, with testing conducted for each permutation as outlined above. These tests were performed within the two population subgroups (Europeans and South Asians) and the results from both subgroups were meta-analysed using the PLINK/SEQ SMP method[62], to control for population stratification.

We tested for an enrichment of rare variants in nine biologically relevant genesets (Supplementary Table 3) and across all available KEGG pathways ($N = 186$) (http://software.broadinstitute.org/gsea/msigdb) using the SMP algorithm[62]. We did this across each of the AF thresholds and functional categories defined above. The BURDEN test was applied across each gene in our case–control population subgroups and its 20,000 permuted datasets. The test-statistic of a given geneset was then defined as the sum of the single-gene statistics for all genes contained in that geneset. To better control for potential baseline differences (such as depth) between cases and controls, the enrichment of each geneset was expressed relative to the enrichment observed exome-wide (when considering all genes harboring at least one rare variant in samples).

**Genome-wide genotyping of VEO-IBD cases**. VEO-IBD cases were genotyped using the Illumina Infinium Core Exome v12.1 chip. Poorly genotyped SNPs, defined as sites with a missing genotype rate >5%[63], were removed. Five European-descent VEO-IBD samples with autosomal heterozygosity rates >3 s.d. from the mean were excluded. Data were phased using SHAPEIT v2[64] and imputation performed using IMPUTE v2[65] with 5010 haplotypes from the 1KG phase 3 data[51]. Genotypes with an INFO score below 0.9 were excluded from the dataset.

**Polygenic risk score calculation**. We assessed the contribution of IBD loci associated with adult-onset IBD to VEO-IBD[21,22]. This analysis made use of the genome-wide genotype and imputation data, rather than our exome-sequences, because the vast majority of IBD associations identified via GWAS are located in intergenic or intragenic regions and are best tagged by non-coding variants. To

enable the VEO-IBD risk scores to be compared with those from non VEO-IBD cases and controls, we obtained high quality genome-wide genotype data for a large European-descent cohort of 13,896 IBD cases from the UK IBD Genetics Consortium[22,23] and 18,843 population controls drawn from The 1958 British Birth Cohort and the UK Blood Service (both genotyped as part of the Wellcome Trust Case Control Consortium project) and the UK Household Longitudinal Study. To minimize population stratification we restricted our risk score analysis to European-descent VEO-IBD cases of high genotype quality ($N = 99$). The SNP set for the risk score calculation was generated by selecting the most associated SNP showing at least suggestive evidence of association ($P < 1 \times 10^{-5}$) with CD or UC in the European cohort of the largest IBD GWAS meta-analysis at the time of the analysis[21]. This resulted in 174 and 146 alleles associated with CD or UC, respectively. Proxy markers with an $r^2 > 0.9$ were selected substitute SNPs that were absent from the VEO-IBD and/or UKIBDGC genotype data. Post QC, a total of 147 CD and 119 UC-associated alleles were available in both the VEO-IBD and the UKIBDGC genotype data. UKIBDGC samples with outlying missingness rates, across all the backbone SNPs used for the risk analysis, were excluded from the analysis. This results in a total of 7578 CD cases (mean age at disease diagnosis = 27 years) and 6318 UC cases (mean age at disease diagnosis of 36 years) available for the analysis. The additive multi-SNP polygenic risk score for each sample was calculated using PLINK2 (www.cog-genomics.org/plink/2.0/), by summing the log of the odds ratio (OR) for each risk allele $j$ carried across all disease loci ($m$):

$$\log(\text{risk}) = \sum_{j=1}^{m} \log(\text{OR}_j) \times x_{ij},$$

where $x_{ij}$ is the number of risk alleles at SNP$_j$ carried by individual$_i$, and OR$_j$ is the allelic odds ratio at SNP$_i$ as estimated by Liu et al. or De Lange et al.[21,22] Risk scores were compared between groups using the Student's $t$-test and assuming a significance threshold of $P < 8 \times 10^{-3}$, which accounts for the six pairwise comparisons between VEO-IBD cases, IBD cases and controls, in CD and UC. Power estimations were performed using the R package pwr. Cohen's $d$ was estimated using the R package lsr.

**Validation of polygenic risk scores in VEO-IBD**. To validate the risk scores obtained in our VEO-IBD COLORS cohort, we used a cohort of 117 VEO-IBD patients from the SickKids Toronto IBD cohort, and a cohort of population controls from the NIDDK IBD Genetics Consortium. The SickKids Toronto IBD cohort comprises 956 samples genotyped using the Immunochip, an Illumina Infinium microarray comprising 196,524 SNPs and small indel markers selected based on results from genome-wide association studies of 12 different immune-mediated diseases. The NIDDK IBD Genetics Consortium control cohorts included 1008 and 2463 controls genotyped with Illumina's Infinium Global Screening Array, which includes a set of fixed markers ($N = 665$k) as well as a set of additional custom markers (NIDDK-Broad, $N = 700,078$; NIDDK-Feinstein, $N = 710,468$). Quality control, genome-wide SNP imputation and the construction of the polygenic risk scores in the SickKids Toronto IBD and NIDDK cohorts was performed using the same pipeline applied to the VEO-IBD COLORS cohort. Briefly, we excluded samples and common variants with missingness >5%. Low frequency variants (MAF < 1%) with missingness >2%, and variants deviated from Hardy-Weinberg equilibrium ($P < 10^{-8}$, Fisher's exact test) were also removed. We also excluded samples with autosomal heterozygosity rates > 3 s.d. from the mean, duplicate samples or close relatives (pi_hat ≥ 0.125), and non European-descent samples identified through PCA. After SNP and sample quality control, the replication cohort consisted of 671 European ancestry cases with genotype data for 147,575 autosomal SNPs, plus 849 NIDDK-Broad and 1754 NIDDK-Feinstein controls with genotype data for 521,576 and 550,555 autosomal SNPs, respectively. Genotypes were pre-phased with SHAPEIT2 v2[64] and imputed with IMPUTE2[65] using the 1KG phase 3 as a reference panel[51]. Imputed data for rs564349 was not available in 1KG Phase 3, so was generated using 1KG Phase 1[53] as a reference panel for the NIDDK control cohorts. The subset of 117 VEO-IBD samples were selected for inclusion in our replication study (detailed characteristics of this VEO subset are provided in Supplementary Table 4). Polygenic risk scores were calculated using the same set of 147 CD and 119 UC associated SNPs as for the COLORS VEO-IBD and UKIBDGC discovery cohorts.

**DHR FACS to measure NADPH activity**. The neutrophil oxidative burst assay to detect reactive oxygen species by DHR FACS assay was performed using standard techniques[66]. Briefly, EDTA blood was incubated with DHR-123 (Life Technologies, D23806) at 2.5 μg/mL for 15 min at 37 °C followed by PMA (Sigma, P1585) at 100 ng/mL stimulation and FACS staining. DHR response was measured in FSC/SCC gated neutrophils. The stimulation index refers to the ratio of the mean fluorescence of the stimulated cells to the mean fluorescence observed in the unstimulated cells in the DHR assay.

**FACS of cells with differential NADPH oxidase activity**. For cell sorting, DHR FACS was performed as described above with a lower PMA stimulation (50 ng/mL, ThermoFisher) to ensure viability for neutrophils in the subsequent gentamicin protection assay. Neutrophils were gated based on cell size and sorted for DHR fluorescence. DHR-high cells were sorted from a healthy control, the neutrophils

from the patient were sorted into a DHR-low and DHR-high population (BD FACS Aria III). Sorting performed with an 85 micron nozzle.

**MiSeq analysis of flow-sorted neutrophils**. The proportion of the flow-sorted cells carrying the mutation was determined by NGS in a targeted 3-PCR approach. The primers were designed using the Eurofins Genomics PCR Primer Design Tool or provided by Illumina Inc. and synthesized by Sigma-Aldrich (Suffolk, United Kingdom). The PlatinumTaq DNA Polymerase High Fidelity Kit (11304-011, Invitrogen) was used according to the instructions of the manufacturer, with a final reaction volume of 10 μL, final primer concentration of 0.2 μM, 25 cycles and an annealing temperature ($T_a$) of 60 °C, unless otherwise stated. A successful PCR was confirmed after each reaction by detecting an appropriately sized band on a 2% agarose gel. The first primer pair (F:ACTCACCCTTTCAAAACCATC, R: ACTTGGCCTTGACCCTTAC) amplified a 513 bp fragment surrounding the variant of interest from 6 ng of genomic DNA. The second primer pair (F: ACACTCTTTCCCTACACGACGCTCTTCCGATCTCACCCTTTTACACTGAC ATCC, R:TCGGCATTCCTGCTGAACCGCTCTTCCGATCTAGTGCCATTTT TCCTGAACTC) used 0.5 μL of the resulting amplicon to amplify a 258 bp fragment composed of 192 bp of the template surrounded by the Illumina Adapter 5′ and 3′ overhangs. This amplicon was diluted 1:100 in nuclease-free water, and 0.5 μL of the dilution was used as a template to append indexes in a third PCR ($T_a$ = 70°). All samples shared a forward primer (AATGATACGGCGACCAC CGAGATCTACACTATAGCCTACACTCTTTCCCTACACGACGCTCTTCCG ATCT) with the 8 bp index TATAGCCT, whilst each had a different reverse primer composed of a common core (CAAGCAGAAGACGGCATACGAGAT--------GA GATCGGTCTCGGCATTCCTGCTGAACCGCTCTTCCGATC) and a unique 8 bp index (--------: AACGTGAT, AAACATCG, ATGCCTAA, AGTGGTCA, ACCACTGT, ACATTGGC, CAGATCTG). The indexed samples were pooled together (2 μL each) and purified using AMPureXP beads (A63880, Beckman Colter) at a ratio of 0.6×. The purified library was quantified with the KAPA Library Quant Kit (Illumina) (KK4824, KAPA Biosystems) on a StepOnePlus Real-Time PCR System, and the size of the amplicon confirmed with a DNA 1000 Kit (5067-1504, Agilent Technologies) on an Agilent 2100 Bioanalyzer System, both according to the instructions of their respective manufacturer. The library was spiked with 30% of a PhiX library provided in the MiSeq v2 Reagent Kit 300 Cycle PE Kit (15033642, Illumina) and sequenced on a MiSeq platform using 150 paired end reads. The resulting reads were filtered against low quality reads, aligned to the reference genome hg19 and the proportions of each allele quantified using the DNA Amplicon (Illumina, Inc) workflow provided on the BaseSpace platform.

**Mosaicism analysis**. The Isohelix swab pack (SK-1S, Isohelix) was used to collect buccal swabs from the patient. DNA was extracted using the QIAmp DNA mini kit (QIAGEN) according to the manufacturer's protocol. For hair follicle analysis, material was treated with 20 μL 1 M DTT, 300 μL buffer ATL and 20 μL proteinase K for 1 h at 56 °C. We then proceeded with the DNeasy blood and tissue kit (Qiagen). PBMCs were isolated from whole blood using Lymphoprep (Axis-Shieldand) and Ficoll gradient centrifugation. Cells were re-suspended in RPMI-1640 (Sigma), the granulocyte layer extracted[67], erythrocytes lysed with water and the white granulocyte pellet was collected. Cells were stained using CD56 (BV510, clone HCD56, Biolegend, catalog number 318340), CD14 (BV650, clone M5E2, Biolegend, catalog number 563420), CD19 (BV711, clone SJ25C1, BD Horizon, catalog number 563038), CD3 (PE/Dazzle 594, clone UCHT1, Biolegend, catalog number 300450), CD4 (BV605, clone OKT4, Biolegend, catalog number 317438), CD8 (AF700, clone SK1, Biolegend, catalog number 344724) and DAPI (1:8000 dilution, Merck, catalog number D9542). To stain the granulocytes, CD16 (PE-Cy7, clone 3G8, Biolegend, catalog number 302016), Siglec-8 (PE, clone 7C9, Biolegend, catalog number 347104) and DAPI (1:8000 dilution, Merck, catalog number D9542) were used. All antibodies were used at a 1:100 dilution unless otherwise stated. Cells were washed and filtered prior to sorting on BD FACS Aria III.

As a control for our sequencing analysis, we extracted DNA from the HEK293T (ATCC-CRL-11268) cells using the DNeasy blood & tissue kit (Qiagen).

Primers for PCR were designed using NCBI Primer Blast for regions flanking the locus of interest and synthesized by Life Technologies. Fifty nanogram DNA amplified using the Phusion High Fidelity DNA Polymerase kit and (#M0530L, New England Biolabs) 5× Phusion HF buffer (New England Biolabs) with a final concentration of 0.5 μM of forward (5′ AAGTGCCCAAAGGTGTCCAA 3′) and reverse (5′ AGCTTCAGATTGGTGGCGTT 3′) primers (CYBB-F and CYBB-R primers, respectively) resulting in a 230 bp fragment in each reaction. A 230 bp fragment containing the mutation site was separated using agarose gel electrophoresis and Sanger sequenced. To quantify the proportion of cells carrying the mutation, we used targeted next generation sequencing of 3 replicates per cell type. Briefly, 5 ng of DNA template was amplified using Phusion® High-Fidelity PCR Master Mix in the presence of EvaGreen Dye (Biotium, USA) followed by thermal cycling in a real-time thermal cycler using a standard PCR protocol. Presence of correct PCR products was confirmed using melting curves. The PCR products were then directly used for five additional cycles of PCR using Phusion DNA polymerase and indexing oligos. Different indices were used for each replicate of all samples. All products were then pooled and purified using AMPure XP beads (Beckman Colter, USA) at a 0.8× bead ratio. The library was sequenced on an Illumina MiSeq platform. Reads were mapped to the hg19 human reference

genome using Bowtie2[68] and visualized using IGV 2.3 (http://software. broadinstitute.org/software/igv/). Variant calling across the region (chrX:37639270-37672714) was carried out using Platypus[69].

**Gentamicin protection assay**. The neutrophil gentamicin protection assay was adapted from Riffelmacher et al, 2017[70]. Briefly, neutrophils were sorted into RPMI1640 (Sigma) and 10% FCS (Sigma). Monocyte derived macrophages were generated from PBMCs as previously described[31]. Forty-thousand neutrophils or macrophages were infected in a 96-well round-bottom plate or flat-bottom plate, respectively, at a MOI 1:10 using Salmonella enterica serovar Typhimurium for 45 min or 1 h, respectively. Cells were treated with 100 μg/mL gentamicin (Sigma) for a further 45 min for neutrophils, or 2 h for macrophages. Cells were then lysed in 1%-Triton X-100 (Sigma) in H2O. Lysates were plated on LB agar plates using the track method and CFU were quantified the following day.

**MDP induced intracellular TNF response**. Quantification of intracellular TNF in monocytes was performed as previously described[31,71]. Briefly, freshly isolated PBMC were rested overnight in RPMI 10% FCS. Unstimulated cells were compared with MDP and LPS (lipopolysaccharide) stimulated cells. Intracellular TNF (MAb11, eBioscience) was detected in CD14+HLADR+monocytes (CD14 (M5E2) and HLA-DR (L243), both BioLegend). Viable cells were detected based on live-dead staining (Fixable Viability Dye, eBioscience).

**SAP expression**. FACS was used to quantify SAP protein expression encoded by *SH2D1A* as previously described[72]. In brief, blood cells were incubated with mouse anti-human CD3 (BD Bioscience, Clone SK7, cat 345767), mouse anti-human CD8-PerCP (BD Bioscience, Clone SK1, cat 345774), and mouse anti-human CD56-PE (BD Bioscience, Clone MY31, cat 345810). Samples were then fixed, washed, an permeabilized. The anti-SAP antibody (Stratech Scientific Biosciences; clone 1C9, cat H00004068-MO1) or isotype control antibody (IgG1 isotype control; BD Biosciences 349040). Samples were again washed and stained with anti-mouse IgG1-FITC (Dako; F0479) before FACS analysis.

**Reporting summary**. Further information on research design is available in the Nature Research Reporting Summary linked to this article.

## Data availability
Sequencing and genotyping data that supports this study have been deposited to the European Genome-phenome Archive (EGA) under the accession code EGAS00001000513 and EGAS00001000924, respectively. All other data are contained in the paper and its supplementary information or available upon request.

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

## Acknowledgements

We thank all individuals who contributed samples to the study. We acknowledge Satish Keshav who sadly died during the revision process of this manuscript and who was instrumental to setup the Oxford IBD cohort study. The COLORS in IBD project was supported by an ESPGHAN collaboration network grant and Crohn's and Colitis UK (M/11/01). The genotyping and sequencing of the COLORS in IBD cohort was funded by the Wellcome Trust (098051; 093885/Z/10/Z). Adult-onset IBD case collections were supported by Crohn's and Colitis UK and sequencing was co-funded by the Wellcome Trust (098051) and the Medical Research Council, UK (MR/J00314X/1). We thank the investigators of the NIDDK IBD Genetics Consortium for sharing previously unpublished genotype data for a large cohort of population controls, used in our polygenic risk analyses. We acknowledge support of the Wellcome Trust (E.G.S., L.M., L.F., J.M., J.C.B. and C.A.A.: 098051), the Crohn's & Colitis Foundation of America (H.H.U.), the Leona M. and Harry B. Helmsley Charitable Trust (H.H.U., A.M., S.S.), the Deutsche Forschungsgemeinschaft (T.S.), the Swiss National Science Foundation (C.P.B.), the Medical Research Council (A.A., T.A.F., DCW), the Medical Research Foundation (D.C.W.) and the Ovarian Cancer Action (A.A.). T.A.B. is supported by a Radcliffe Department of Medicine/MRC Scholars Program Studentship. A.M.M. is funded by a CIHR Foundation Grant and the Tier 1 Canada Research Chair in Pediatric IBD. The NIDDK IBD Genetics Consortium is supported by National Institute of Diabetes and Digestive and Kidney Diseases grants. We acknowledge the contribution of the Oxford Biomedical Research Center, which is supported by the National Institute for Health Research. We thank the INTERVAL Study for contributing sequencing data. INTERVAL was supported by: NHSBT, NIHR, NIHR BioResource, NIHR [Cambridge Biomedical Research Center at the Cambridge University Hospitals NHS Foundation Trust] [∗], NIHR BTRU in Donor Health and Genomics, UK MRC, BHF, Wellcome Trust, and the ESRC. J.D. is funded by the NIHR [Senior Investigator Award]. ∗The views expressed are those of the authors and not necessarily those of the NHS, the NIHR or the Department of Health and Social Care.

## Author contributions

E.G.S., L.M., L.F., R.W., H.H.U. and C.A.A. analysed the sequence and genotype data. T.S., S.P., A.C., K.G., T.B., T.F., E.M.C., J.C.M. and A.A. performed functional validation experiments. E.C.S., T.S., J.K., N.C., C.P., A.R., R.R., F.B., M.A., R.H., M.Z., K.F., C.B., S.T., J.S., M.P., N.T., A.W., C.M., J.S., P.S., W.O., D.J.R., J.D., J.B., A.G., A.M., N.S., A.E., S.S., D.W. and H.H.U. contributed to recruitment of patient or control groups. E.G.S., H.H.U. and C.A.A. wrote the manuscript. All authors read and approved the final version of the manuscript. H.H.U. and C.A.A. jointly supervised and coordinated the project.

## Competing interests

The authors have made the following disclosures: C.A.A. has received consultancy fees from Genomics plc and Illumina. H.H.U. received collaborative research support or consultancy fees from Eli Lilly, UCB Pharma, Celgene, Regeneron, Boehringer Ingelheim, Pfizer, and AbbVie. SPT has been adviser to, in receipt of educational or research grants from, or invited lecturer for AbbVie, Amgen, Asahi, Biogen, Boehringer Ingelheim, BMS, Cosmo, Elan, Enterome, Ferring, FPRT Bio, Genentech/Roche, Genzyme, Glenmark, GW Pharmaceuticals, Immunocore, Immunometabolism, Janssen, Johnson & Johnson, Lilly, Merck, Novartis, Novo Nordisk, Ocera, Pfizer, Shire, Santarus, SigmoidPharma, Synthon, Takeda, Tillotts, Topivert, Trino Therapeutics with Wellcome Trust, UCB Pharma, Vertex, VHsquared, Vifor, Warner Chilcott, and Zeria. The remaining authors declare no competing interests.

## Additional information

Eva Gonçalves Serra[1], Tobias Schwerd[2,109], Loukas Moutsianas[1], Athena Cavounidis[2], Laura Fachal[1], Sumeet Pandey[2], Jochen Kammermeier[3], Nicholas M. Croft[4,5], Carsten Posovszky[6], Astor Rodrigues[7], Richard K. Russell[8], Farah Barakat[4,5], Marcus K.H. Auth[9], Robert Heuschkel[10], Matthias Zilbauer[10], Krzysztof Fyderek[11], Christian Braegger[12], Simon P. Travis[2], Jack Satsangi[2,13], Miles Parkes[14], Nikhil Thapar[3], Helen Ferry[2], Julie C. Matte[1], Kimberly C. Gilmour[3], Andrzej Wedrychowicz[11], Peter Sullivan[7], Carmel Moore[15,16], Jennifer Sambrook[16,17], Willem Ouwehand[1,15,16,17], David Roberts[15,18,19], John Danesh[1,16], Toni A. Baeumler[20], Tudor A. Fulga[20], Eli M. Carrami[20], Ahmed Ahmed[20,21], Rachel Wilson[2], Jeffrey C. Barrett[1], Abdul Elkadri[22,23], Anne M. Griffiths[22,23], COLORS in IBD group investigators*, Oxford IBD cohort study investigators*, INTERVAL Study*, Swiss IBD cohort investigators*, UK IBD Genetics Consortium*, NIDDK IBD Genetics Consortium*, Scott B. Snapper[24,25,26], Neil Shah[3], Aleixo M. Muise[22,23], David C. Wilson[27], Holm H. Uhlig[2,7,110✉] & Carl A. Anderson[1,110✉]

[1]Wellcome Sanger Institute, Wellcome Genome Campus, Hinxton, UK. [2]Translational Gastroenterology Unit, University of Oxford, Oxford, UK. [3]Great Ormond Street Hospital, London, UK. [4]Blizard Institute, Barts and the London School of Medicine, Queen Mary University of London,

London, UK. [5]The Royal London Children's Hospital, Barts Health NHS Trust, London, UK. [6]Universitätsklinikum, Ulm, Germany. [7]Department of Paediatrics, University of Oxford, Oxford, UK. [8]Royal Hospital for Children, Glasgow, UK. [9]Alder Hey Children's Hospital, Liverpool, UK. [10]Addenbrooke's Hospital, Cambridge, UK. [11]Department of Paediatrics, Gastroenterology and Nutrition, Jagiellonian University Medical College, Krakow, Poland. [12]Division of Gastroenterology and Nutrition and Children's Research Center, University Children's Hospital Zurich, Zurich, Switzerland. [13]Institute of Genetics and Molecular Medicine, University of Edinburgh, Scotland, UK. [14]IBD Research Unit, Department of Gastroenterology, Addenbrooke's Hospital, Cambridge, UK. [15]NIHR Blood and Transplant Research Unit in Donor Health and Genomics, Department of Public Health and Primary Care, University of Cambridge, Cambridge, UK. [16]INTERVAL Coordinating Centre, Department of Public Health and Primary Care, University of Cambridge, Cambridge, UK. [17]Department of Haematology, University of Cambridge, Cambridge, UK. [18]NHS Blood and Transplant - Oxford Centre, Level 2, John Radcliffe Hospital, Oxford, UK. [19]Biomedical Research Centre, Oxford – Haematology Theme, Radcliffe Department of Medicine, University of Oxford, John Radcliffe Hospital, Oxford, UK. [20]Weatherall Institute of Molecular Medicine and the Radcliffe Department of Medicine, University of Oxford, John Radcliffe Hospital, Oxford, UK. [21]National Institute of Health Research Oxford Biomedical Research Centre, Surgical Innovation and Evaluation and Molecular Diagnostics Themes, University of Oxford, Oxford, UK. [22]Department of Biochemistry and Pediatrics, Faculty of Medicine, University of Toronto, Toronto, ON, Canada. [23]SickKids Inflammatory Bowel Disease Centre and Cell Biology Program, Research Institute, Hospital for Sick Children, Toronto, ON, Canada. [24]Division of Gastroenterology, Hepatology and Nutrition, Boston Children's Hospital, Boston, MA, USA. [25]Harvard Medical School, Boston, MA, USA. [26]Division of Gastroenterology, Brigham and Women's Hospital, Boston, MA, USA. [27]Child Life and Health, University of Edinburgh, Edinburgh, UK. [109]Present address: Dr. von Hauner Children's Hospital, Department of Pediatrics, University Hospital, Ludwig Maximilians University, Munich, Germany. [110]These authors jointly supervised: Holm H. Uhlig, Carl A. Anderson. *Members of the COLORS in IBD group investigators, Oxford IBD cohort study investigators, INTERVAL Study, Swiss IBD cohort investigators, UK IBD Genetics Consortium, NIDDK IBD Genetics Consortium appears at the end of the paper. ✉email: holm.uhlig@ndm.ox.ac.uk; carl.anderson@sanger.com

## COLORS in IBD group investigators

Marlen Zurek[28], Caterina Strisciuglio[29], Mamoun Elawad[30] & Bernice Lo[30]

[28]St. Georg Hospital, Leipzig, Germany. [29]Department of Translational Medical Sciences, Section of Pediatrics, University of Naples, Naples, Italy. [30]Sidra Medical and Research Center, Doha, Qatar

## Oxford IBD cohort study investigators

Carolina Arancibia-Carcamo[2], Adam Bailey[2], Ellie Barnes[2], Elizabeth Louise Bird-Lieberman[2], Oliver Brain[2], Barbara Braden[2], Jane Collier[2], James East[2], Lucy Howarth[7], Satish Keshav[2], Paul Klenerman[2], Simon Leedham[2], Rebecca Palmer[2], Fiona Powrie[2] & Alison Simmons[2]

## INTERVAL Study

Matthew Walker[15], Zoe Tolkien[15], Stephen Kaptoge[15], David Allen[18], Susan Mehenny[31], Jonathan Mant[15], Emanuele Di Angelantonio[15] & Simon G. Thompson[15]

[31]NHS Blood and Transplant, Longley Lane, Sheffield, UK

## Swiss IBD cohort investigators

Bahtiyar Yilmaz[32,33], Pascal Juillerat[32,33], Markus Geuking[32], Reiner Wiest[32], Andrew J. Macpherson[32,33], Francisco Damian Bravo[33], Lukas Brügger[33], Ove Carstens[33], Ulrike Graf Bigler[33], Benjamin Heimgartner[33], Monica Rusticeanu[33], Sybille Schmid (-Uebelhart)[33], Bruno Strebel[33], Aurora Tatu[33], Radu Tutuian[33], Reiner Wiest[33], Ove Øyås[34], Charlotte Ramon[34], Jörg Stelling[34], Yannick Franc[35], Nicolas Fournier[35], Valerie E.H. Pittet[35], Bernard Burnand[35], Mara Egger[35], Yannick Franc[35], Delphine Golay[35], Astrid Marot[35], Leilla Musso[35], Valérie Pittet[35], Jean-Benoît Rossel[35], Vivianne Seematter[35], Joachim Sommer[35], Rachel Vulliamy[35], Pierre Michetti[36,37], Michel H. Maillard[36,37], Céline Keller[36], Michel H. Maillard[36,37], Andreas Nydegger[36,37], Alain Schoepfe[36], Eva Archanioti[37], Jessica Ezri[37], Montserrat Fraga[37], Alain Schoepfe[37], Christoph Müller[38], Gerhard Rogler[39], Luc Biedermann[39], Mirjam Blattmann[39], Sabine Burk[39], Barbara Dora[39], Michael Fried[39], Benjamin Misselwitz[39], Beat Müllhaupt[39], Nicole Obialo[39], Daniel Pohl[39], Nadia Raschle[39], Gerhard Rogler[39], Michael Scharl[39], Stephan Vavricka[39], Roland Von Känel[39], Jonas Zeitz[39], Karim Abdelrahman[40], Gentiana Ademi[41], Jan Borovicka[41], Stephan Brand[41], Remus Frei[41], Johannes Haarer[41], Christina Knellwolf (-Grieger)[41], Claudia Krieger(-Grübel)[41], Patrizia Künzler[41], Christa Meyenberger[41],

Pamela Meyer[41], Nina Röhrich[41], Mikael Sawatzki[41], Martin Schelling[41], Gian-Marco Semadeni[41], Michael Sulz[41], Dorothee Zimmermann[41], Patrick Aepli[42], Dominique H. Criblez[42], Cyrill Hess[42], Jean-Pierre Richterich[42], Johannes Spalinger[42], Dominic Staudenmann[42], Andreas Stulz[42], Stefanie Wöhrle[42], Amman Thomas[43], Claudia Anderegg[44], Henrik Köhler[44], Rachel Kusche[44], Anca-Teodora Antonino[45], Eviano Arrigoni[46], José M. Bengoa[46], Sophie Cunningham[46], Philippe de Saussure[46], Laurent Girard[46], Diana Bakker de Jong[47], Polat Bastürk[47], Simon Brunner[47], Lukas Degen[47], Petr Hruz[47], Carolina Khalid-de Bakker[47], Jan Niess[47], Bruno Balsiger[48], Janine Haldemann[48], Gaby Saner[48], Frank Seibold[48], Peter Bauerfeind[49], Andrea Becocci[50], Dominique Belli[50], Janek Binek[51], Peter Hengstler[51], Stephan Boehm[52], Tujana Boldanov[53], Patrick Bühr[12], Rebekka Koller[12], Vanessa Rueger[12], Arne Senning[12], Emanuel Burri[54], Sophie Buyse[55], Dahlia-Thao Cao[56], Fabrizia D'Angelo[57], Joakim Delarive[58], Christopher Doerig[59], Roxane Hessler[59], Claudia Preissler[60], Ronald Rentsch[61], Branislav Risti[62], Marc Alain Ritz[63], Michael Steuerwald[63], Jürg Vögtlin[63], Markus Sagmeister[64], Bernhard Sauter[65], Susanne Schibli[66], Christiane Sokollik[66], Johannes Spalinger[66], Hugo Schlauri[67], Jean-François Schnegg[68], Mariam Seirafi[69], Holger Spangenberger[70], Philippe Stadler[71], Peter Staub[72], Volker Stenz[73], Michela Tempia-Caliera[74], Joël Thorens[75], Kaspar Truninger[76], Patrick Urfer[77], Francesco Viani[78], Dominique Vouillamoz[79], Silvan Zander[80] & Tina Wyli[81]

[32]Maurice Müller Laboratories, Department for Biomedical Research, University of Bern, Bern, Switzerland. [33]Department of Visceral Surgery and Medicine, Bern University Hospital, University of Bern, Bern, Switzerland. [34]Department of Biosystems Science and Engineering and SIB Swiss Institute of Bioinformatics, ETH Zurich, Basel, Switzerland. [35]Institute of Social and Preventive Medicine (IUMSP), Lausanne University Hospital, Lausanne, Switzerland. [36]Gastroenterology La Source-Beaulieu, Lausanne, Switzerland. [37]Service of Gastroenterology and Hepatology, Department of Medicine, Centre Hospitalier Universitaire Vaudois and University of Lausanne, Lausanne, Switzerland. [38]Division of Experimental Pathology, Institute of Pathology, University of Bern, Bern, Switzerland. [39]Department of Gastroenterology and Hepatology, University Hospital Zurich, University of Zurich, Zurich, Switzerland. [40]Clinique de Montchoisi, Lausanne, Switzerland. [41]Kantonsspital St-Gallen, St. Gallen, Switzerland. [42]Kantonsspital Luzern, Luzern, Switzerland. [43]GI private practice, Waldkirch, St. Gallen, Switzerland. [44]Kantonspital Aarau, Klinik für Kinder und Jugendliche, Aarau, Switzerland. [45]Hôpital Riviera—Site du Samaritain, Vevey, Vaud, Switzerland. [46]GI private practice, Geneva, Switzerland. [47]Department of Gastroenterology and Hepatology, Basel University Hospital, Basel, Switzerland. [48]Gastroenterologische Praxis, Bern, Switzerland. [49]Department Gastroenterology and Hepatology, Stadtspital Triemli, Zurich, Switzerland. [50]Department of Pediatric, Geneva University Hospital, Geneva, Switzerland. [51]Gastroenterologie am Rosenberg, St. Gallen, Switzerland. [52]Spital Bülach, Bülach, Zurich, Switzerland. [53]Department of Biomedicine, University of Basel, Basel, Switzerland. [54]Department Gastroenterology, Kantonsspital Liestal, Liestal, Switzerland. [55]GI private practice, Yverdon-les-Bains, Liestal, Switzerland. [56]Hôpital Neuchâtelois, La Chaux-de-fonds, Neuchâtel, Switzerland. [57]Department Gastroenterology and Hepatology, Geneva University Hospital, Geneva, Switzerland. [58]GI private practice, Lausanne, Switzerland. [59]Clinique Cecil, Lausanne, Switzerland. [60]Kantonsspital Olten, Olten, Switzerland. [61]GI Private Practice, St. Gallen, Switzerland. [62]GI Practice, Dietikon, Switzerland. [63]GI Practice, Liestal, Switzerland. [64]GI Private Practice, Heerbrugg, Switzerland. [65]Klinik Hirslanden Zürich, Zurich, Switzerland. [66]Kinderklinik Bern, Bern University Hospital, Bern, Switzerland. [67]Derby Center, Wil, Switzerland. [68]GI Private Practice, Montreux, Switzerland. [69]Clinique La Colline, Geneva, Switzerland. [70]Kantonsspital Wolhusen, Wolhusen, Switzerland. [71]GI Private Practice, Payerne, Switzerland. [72]Spital Heiden Appenzell Ausserrhoden, Heiden, Switzerland. [73]Kantonsspital Münsterlingen, Münsterlingen, Switzerland. [74]Clinique des Grangettes, Chêne-Bougeries, Switzerland. [75]GI Private Practice, Yverdon, Switzerland. [76]GI Private Practice, Langenthal, Switzerland. [77]Hirslanden Klinik Aarau, Gastro Zentrum, Aarau, Switzerland. [78]Private Practice, Vevey, Switzerland. [79]Private Practice, Pully, Switzerland. [80]Spital Limmattal, Schlieren, Switzerland. [81]Infirmière de Recherche chez CHUV Lausanne University Hospital, Lausanne, Switzerland

## UK IBD Genetics Consortium

L. Jostins[82], N.A. Kennedy[83], T. Ahmad[83], C.A. Lamb[84], C. Edwards[85], A. Hart[86], C. Hawkey[87], J.C. Mansfield[88], C. Mowat[89], W.G. Newman[90], A. Simmons[2], M. Tremelling[91], J.C. Lee[14], N.J. Prescott[92], C.G. Mathew[92] & C.W. Lees[93]

[82]The Wellcome Trust Centre for Human Genetics, Oxford, UK. [83]Precision Medicine Exeter, University of Exeter, Exeter, UK. [84]Institute of Cellular Medicine, Newcastle University, Newcastle Upon Tyne, UK. [85]Department of Gastroenterology, Torbay Hospital, Torbay, UK. [86]Department of Medicine, St. Mark's Hospital, Harrow, UK. [87]Nottingham Digestive Diseases Centre, Queens Medical Centre, Nottingham, UK. [88]Institute of Human Genetics, Newcastle University, Newcastle upon Tyne, UK. [89]Department of Medicine, Ninewells Hospital and Medical School, Dundee, UK. [90]Genetic Medicine, Manchester Academic Health Science Centre, Manchester, UK. [91]Gastroenterology & General Medicine, Norfolk and Norwich University Hospital, Norwich, UK. [92]Department of Medical and Molecular Genetics, King's College London School of Medicine, Guy's Hospital London, London, UK. [93]Gastrointestinal Unit, Western General Hospital University of Edinburgh, Edinburgh, UK

## NIDDK IBD Genetics Consortium

D.P.B. McGovern[94], S.R. Targan[94], G. Botwin[94], E. Mengesha[94], P. Fleshner[94], C. Landers[94], D. Li[94], J.D. Rioux[95], A. Bitton[96], J. Côté-Daigneault[97], M.J. Daly[98], R. Xavier[99], K. Morris[96], G. Boucher[95], J.H. Cho[100], C. Abraham[101], M. Merad[102], B. Sands[103], I. Peter[103], K. Hao[51], Y. Itan[100], R.H. Duerr[104], L. Konnikova[104], M.B. Schwartz[104], S. Proksell[104], E. Johnston[104], V. Miladinova[104], W. Chen[104], S.R. Brant[105], L. Datta[105], M.S. Silverberg[106], L.P. Schumm[107], S. Birch[107], M. Giri[100], K. Gettler[100], Y. Sharma[100], C. Stevens[99], M. Lazarev[108] & T. Haritunians[94]

[94]F. Widjaja Inflammatory Bowel and Immunobiology Research Institute, Cedars-Sinai Medical Center, Los Angeles, CA, USA. [95]Montreal Heart Institute, Research Center, Montreal, Canada. [96]McGill University Health Centre, Montreal, Quebec, Canada. [97]Centre Hospitalier de l'Université de Montréal, Montreal, Canada. [98]Institute of Molecular Medicine Finland, Helsinki, Finland. [99]Broad Institute of MIT, Cambridge, Massachusetts, USA. [100]Charles Bronfman Institute for Personalized Medicine, Departments of Medicine and Genetics, Icahn School of Medicine, New York, USA. [101]Department of Medicine, Yale University, New Heaven, USA. [102]Precision Immunology Institute, Icahn School of Medicine at Mount Sinai, New York, USA. [103]Department of Medicine, Icahn School of Medicine at Mount Sinai, New York, USA. [104]Department of Medicine, University of Pittsburgh School of Medicine, Pittsburgh, PA, USA. [105]Department of Medicine, Rutgers University, New Brunswick, USA. [106]Samuel Lunenfeld-Tanenbaum Research Institute, Mount Sinai Hospital, University of Toronto, New York, USA. [107]Department of Public Health Sciences, University of Chicago, Chicago, USA. [108]Department of Medicine, Johns Hopkins University School of Medicine, Baltimore, MD, USA

