## [Peer Review File · Nature Communications]

Reviewers' Comments:

Reviewer #1:

Remarks to the Author:

The authors started off with an attempt to identify previously undiscovered rare variants that contribute to monogenic VEO-IBD by performing whole exome sequencing with a particular focus on previously reported monogenic IBD and/or IBD-like inflammatory phenotypic genes, and ended up finding/presenting four novel, pathogenic variants in primary immunodeficiency genes, XIAP, SH2D1A, CYBA and CYBB of 4 out of 145 patients with VEO-IBD. Further, their screening for VEO-IBD predisposition genes revealed a mutation in CYBB that is not at the expected heterozygous or homozygous ratios for a germline variant, which is, inferred by the authors, as a somatic mosaic mutation. In addition, authors reported a potential possibility of previously identified adult-onset-IBD associated GWAS variants contributing to VEO-IBD.

Major points:

1) The authors effectively used sequencing and bioinformatic approaches to open several interesting lines of investigation into the pathophysiology driving this phenotype; however, the biological relevance and impact of these findings either falls short of rigor or novelty. The main finding of this study - identification of four pathogenic variants in primary immunodeficiency genes, XIAP, SH2D1A, CYBA and CYBB of 4 patients of VEO-IBD - these genes are previously well recognized for their roles in monogenic VEO-IBD, and other IBD-like inflammatory phenotypes (for instance, Kelsen et al., *Gastroenterology*; 2015). Besides, showing that these 4 novel, extremely rare variants are deleterious/functional by experimental follow-up, the biological relevance of these genes in causing these diseases is very well established by prior studies. Nevertheless, identification of these private deleterious mutations are extremely useful for those particular patients in making treatment choices.

2) The observation that the polygenic risk score computed based on the known IBD-associated common variants (adult-onset) is significantly higher in VEO-IBD cases compared to controls remains primarily descriptive, and is already known in the literature. The authors hypothesized that "the extent of to which the polygenic background influences VEO-IBD susceptibility varies depending on the monogenic foreground", however, although, they stated examples of fully-penetrant IL10 pathway mutations in VEO-IBD and MODY in type 1 diabetes, the rationale behind such hypothesis is not clear based on the current findings reported here. It would be useful to separately compare polygenic risk scores of VEO-IBD patients with and without the identified monogenic variants to controls as a means to support the claimed hypothesis. If possible, in order to increase the number of VEO-IBD patients with monogenic findings, previously published monogenic VEO-IBD data sets (with GWAS data) can be jointly analyzed with the current data set. It would also be useful to add a sentence or two in the discussion section to expand the relevance of this finding of adult-onset-IBD common variants contributing to VEO-IBD.

3) The finding of cellular mosaicism as a novel non-classical mechanism underlying VEO-IBD remains too preliminary. It is not clear, whether the reported somatic mutation in CYBB is post-zygotic mosaic mutation or lineage-restricted somatic mosaic mutation. More discussion on the finding of mosaicism in both immune cells and buccal epithelium would help. And most importantly, separating the FACS sorted cells (for example, from figure 1C), and performing Sanger sequencing to validate the observation of CYBB somatic mutation is highly encouraged.

Reviewer #3:

Remarks to the Author:

Gonçalves Serra and colleagues investigate genetic factors contributing to inflammatory bowel disease (IBD) in a cohort of 145 very-early-onset (VEO) patients (onset <6 years) in whom no Mendelian disorders were clinically suspected. Firstly, they screen for rare, disruptive mutations in

67 genes known to be associated with Mendelian disorders with IBD-like inflammatory phenotypes ascertained from the literature. They identify five patients from four families with mutations in the genes CYBB, CYBA, XIAP and SH2D1A. Three mutations are on the X-chromosome in males of which one is a mosaic de novo mutation affecting 70% of cells in all immune cell subsets. Detailed functional evidence is provided for these mutations. Subsequently, authors calculate a polygenic risk score for known genetic risk factors for CD and UC and demonstrate that the polygenic risk score for CD and UC is significantly increased in the VEO cases compared to healthy controls. They conclude that the polygenic component underlying adult IBD contributes to the VEO cases as well, which is in line with the identification in previous GWAS of risk loci shared between pediatric and adult-onset IBD.

The manuscript is elegantly written and reads very well. Authors took great care in the genetic and functional evaluation of candidate mutations, and evidence for a pathogenic status of the reported variants is strong. Results broaden the range of clinical presentations and phenotypes associated with mutations in these genes. This is of high importance for the clinical care of VEO-IBD patients. Somatic mosaicism for CYBB has been demonstrated previously for the immunodeficiency CGD but the current report adds to the growing support of somatic mutations in autoimmune disease. Limitations of the WES part to a subset of genes and of the design of the polygenic risk score part limit to some extent novelty towards the wider scientific community.

Major comments:

- 1) WES is performed, but analysis is limited to 67 genes already known to be associated with Mendelian disorders with IBD-like inflammatory phenotypes ascertained from the literature. Has evaluation of other genes (eg. all PID genes) been possible, will they be reported elsewhere and/or undertaken as a future perspective?
- 2) In the discussion of disease-causing variants in pediatric IBD, some very recent advance-online publications are missing (e.g. Gastroenterology issues of Feb and March 2018).
- 3) The polygenic risk score for CD and UC is highly significantly increased in VEO-IBD compared to healthy controls. Conclusions that can be drawn from the comparison of VEO versus adult-onset cases are limited as the VEO cases are not subdivided in CD or UC like the adult-onset cases and by the sample size of 99 patients from this single cohort. Similarly, these factors limit the correlation with age at onset.
- 4) In the discussion, authors hypothesize that the extent to which the polygenic background influences VEO-IBD susceptibility varies depending on the monogenic foreground. Their data do not allow addressing this hypothesis, as they indeed indicate.

Minor comments:

- 1) Description of patient cohort: how many patients were excluded from the original cohort because of a clinical diagnosis of a known Mendelian disease was suspected and confirmed? Together with the currently found mutations this would provide an estimate of the Mendelian contribution to VEO-IBD at least in this particular dataset.
- 2) Ancestry of the mutation carriers should be discussed in the Results section as several appear of non-European ancestry.
- 3) A conservative INFO score threshold of 0.9 is set for the polygenic risk score. What is the proportion of missing data resulting from the application of this threshold and how were missing data handled? Is there a difference in missing rate between VEO cases and adult cases and controls as they were obtained from different sources?
- 4) Some references are incomplete.

Reviewers' comments:

Reviewer #1 (Remarks to the Author):

The authors started off with an attempt to identify previously undiscovered rare variants that contribute to monogenic VEO-IBD by performing whole exome sequencing with a particular focus on previously reported monogenic IBD and/or IBD-like inflammatory phenotypic genes, and ended up finding/presenting four novel, pathogenic variants in primary immunodeficiency genes, XIAP, SH2D1A, CYBA and CYBB of 4 out of 145 patients with VEO-IBD. Further, their screening for VEO-IBD predisposition genes revealed a mutation in CYBB that is not at the expected heterozygous or homozygous ratios for a germline variant, which is, inferred by the authors, as a somatic mosaic mutation. In addition, authors reported a potential possibility of previously identified adult-onset-IBD associated GWAS variants contributing to VEO-IBD.

Major points:

1) The authors effectively used sequencing and bioinformatic approaches to open several interesting lines of investigation into the pathophysiology driving this phenotype; however, the biological relevance and impact of these findings either falls short of rigor or novelty. The main finding of this study - identification of four pathogenic variants in primary immunodeficiency genes, XIAP, SH2D1A, CYBA and CYBB of 4 patients of VEO-IBD - these genes are previously well recognized for their roles in monogenic VEO-IBD, and other IBD-like inflammatory phenotypes (for instance, Kelsen et al., *Gastroenterology*; 2015). Besides, showing that these 4 novel, extremely rare variants are deleterious/functional by experimental follow-up, the biological relevance of these genes in causing these diseases is very well established by prior studies. Nevertheless, identification of these private deleterious mutations are extremely useful for those particular patients in making treatment

One of our main findings (as rightly pointed out above), was the identification of pathogenic variants of Mendelian disorders that can present with IBD including primary immunodeficiency (PID) genes (*XIAP*, *SH2D1A*, *CYBB* and *CYBA*). These variants were identified in VEO-IBD patients in whom no Mendelian disorders were clinically suspected. We are aware, and do acknowledge (both in the introduction and discussion of the manuscript), that these genes have been previously identified in VEO-IBD patients. However, our findings broaden the range of clinical presentations and phenotypes associated with mutations in these genes (as pointed out by Reviewer #3). More importantly, these findings suggest that PIDs caused by rare genetic variants can be found in VEO-IBD patients even if no Mendelian disease was clinically suspected, which highlights that genetic screening for such genes is relevant across the entire patient group (and not only in patients with an IBD-like phenotype with a suspected underlying Mendelian cause). This is of high importance for the clinical care of VEO-IBD patients (as pointed out by Reviewer #3) and points the novelty of our work.

We respectfully, but strongly, disagree with the reviewer that our study falls short of rigor. The quality control and association studies we performed on the sequence data were to the highest possible standard, the clinical interpretation of the variants identified through this analysis was in line with the ACMG/AMP criteria, and we conducted detailed functional evaluation of all mutations we causally implicate in VEO-IBD (as pointed out by Reviewer #3). Indeed, we hope our manuscript serves as an example to others in the field of how such studies should be rigorously performed.

Further, the availability of exome sequencing data allowed us to conduct, for the first time, exome-wide gene-based tests, in an attempt to identify novel VEO-IBD associated genes with an increased burden of rare variants in cases versus a large set of controls (N=3,588). Exome sequencing in pediatric IBD cohorts (including some VEO-IBD patients) was conducted in three previous studies, all of which then only looked at a smaller number of genes, including known IBD (N=169; Christodoulou et al 2013), autoimmune (N=33; Andreoletti et al 2015), or PID genes (N=400; Kelsen et al 2015) loci. None of these studies conducted a systematic and comprehensive exome-wide gene-based association tests. To increase statistical power, we next conducted rare-variant burden tests across multiple related genes, such as those that reside in the same biological pathway (now included in the main text following revision, rather than the Supplementary Information). An important result of these analysis was the lack of replication ($P = 0.7$) of a previously reported signal. This means that we could not replicate the by Kelsen et al

2015 reported over-representation of rare, damaging variation (AF<0.1%) in PID-associated genes (N=400) in VEO-IBD patients.

Genome-wide genotyping allowed us to move beyond rare variants and generate polygenic risk scores (PRS) for VEO-IBD patients, based on known IBD risk loci. CD and UC risk scores were significantly greater in VEO-IBD cases compared to population controls. This constituted a **main** finding of our study, and the first time that a polygenic component has been shown to operate in a cohort of exclusively VEO-IBD patients (this point is now clearly noted in the main text).

2) The observation that the polygenic risk score computed based on the known IBD-associated common variants (adult-onset) is significantly higher in VEO-IBD cases compared to controls remains primarily descriptive, and is already known in the literature.

We have been unable to identify any papers in the literature (including pre-submissions in BiorXiv) that describe an excess of common IBD risk variants in cohorts of exclusively VEO-IBD cases. However, if the reviewer can point us towards any such paper, we would be happy to re-interpret our work in light of this.

It has been well-established that adult (age at onset ≥ 17 yrs) and/or paediatric onset IBD (age at onset < 17 yrs) are both complex traits, with thousands of genetic and environmental risk factors likely contributing to an individual's risk of disease. It has also been established that there is a weak, but statistically significant negative relationship between the polygenic risk score for CD/UC and age at diagnosis (Ananthakrishnan et al 2014, Cleynen et al 2015 and Cutler et al 2015). However, all of these studies had very few individuals with VEO-IBD (age at onset < 6 years) so did not exclusively quantify the contribution of polygenic burden to VEO-IBD. This is an important and significant question because we know that, unlike adult or childhood onset IBD, VEO-IBD can be caused by Mendelian diseases driven by single causal variants. Here, we show for the first time that common genetic variation associated with adult onset IBD also plays a role in VEO-IBD. This is an important discovery for understanding the genetic architecture of IBD across age-at-onset groups.

3) The authors hypothesized that "the extent to which the polygenic background influences VEO-IBD susceptibility varies depending on the monogenic foreground", however, although, they stated examples of fully-penetrant IL10 pathway mutations in VEO-IBD and MODY in type 1 diabetes, the rationale behind such hypothesis is not clear based on the current findings reported here. It would be useful to separately compare polygenic risk scores of VEO-IBD patients with and without the identified monogenic variants to controls as a means to support the claimed hypothesis. If possible, in order to increase the number of VEO-IBD patients with monogenic findings, previously published monogenic VEO-IBD data sets (with GWAS data) can be jointly analyzed with the current data set. It would also be useful to add a sentence or two in the discussion section to expand the relevance of this finding of adult-onset-IBD common variants contributing to VEO-IBD.

We apologise for the confusion that our wording in the discussion seems to have caused. We are not claiming that *"the extent to which the polygenic background influences VEO-IBD susceptibility varies depending on the monogenic background"* is a result of our study. As the reviewer points out, our data are insufficient to make this statement. Our goal in the discussion was simply to suggest some potential genetic architectures for VEO-IBD (and provide examples of other diseases consistent with those architectures) and then experiments that would need to be carried out to test if these architectures do indeed play a role in VEO-IBD. Indeed, we clearly state that *"To thoroughly test these models in VEO-IBD would require many hundreds or thousands of VEO-IBD cases (with and without Mendelian diagnoses) to be whole-genome sequenced at high coverage (or exome-sequenced and genotyped using a genome-wide microarray)"*, thus making it clear that further data is needed. We feel that this is an important "setting of the stage" for future work and would like to keep it in the discussion. We have significantly revised the text to make it more clear that we are not claiming these *"genetic architecture models"* have been demonstrated by our current data. Furthermore, we believe it is far outside the scope of this project to undertake this research as part of a paper describing the results of the COLORS in IBD project. Bringing together various disease cohorts of European/Caucasian descent, to allow us to generate and compare risk burden scores,

would be required for a project of this scale; that effort could take years and is very much a project in and of its own right, rather than a “bolt on” to the COLORS in IBD project.

4) The finding of cellular mosaicism as a novel non-classical mechanism underlying VEO-IBD remains too preliminary. It is not clear, whether the reported somatic mutation in CYBB is post-zygotic mosaic mutation or lineage-restricted somatic mosaic mutation. More discussion on the finding of mosaicism in both immune cells and buccal epithelium would help. And most importantly, separating the FACS sorted cells (for example, from figure 1C), and performing Sanger sequencing to validate the observation of CYBB somatic mutation is highly encouraged.

We apologise to the reviewer if our previous description of our work was unclear. We have revised the description and added the experiments suggested by the reviewer.

CYBB is located on the X chromosome and the patient is male. This suggests that the mutation is post-zygotic because a) the patient's mother does not carry the mutation and thus the variant is not inherited (unless the mother herself is mosaic and the mutation in the patient a reversion event, which is extremely unlikely) and b) only 50-70% of cells carry the mutation; if the variant was a de-novo event in the egg that went on to form the zygote, then all of the patient's cells would carry the mutation. Thus, the only option that remains likely is that the mutational event occurred after the formation of the zygote. Since we can detect the pathogenic variant in haematopoietic cells as well as buccal cells and hair follicles, this event is not lineage-restricted and likely occurred somewhere between day days two and five, and certainly before day nine, the time when mesoderm and ectoderm separate. To establish this, we sequenced DNA from several flow cytometry assay (FACS) sorted immune cell subsets, as well as buccal swabs containing epithelial cells and hair follicles from the patient and compared those with wild type cells as a control (Figure 1F-G; see Supplementary Information). In all immune cell subsets of the patient (neutrophils, monocytes, CD4+ T cells, CD20+ B cells, NK cells) we found the mutation was present in around 70% of calls (range: 63-76%), whereas in buccal swabs and hair follicles it was in a similar range but more variable (range: 34-90% of calls contain the pathogenic variant). This indicates that both mesodermal (haematopoietic cells) and ectodermal (cheek epithelia and hair follicles) derived cells are affected and that the mutation arose early in embryogenesis and is not lineage-restricted. We have attempted to improve the clarity of the text describing this work and present the data in **(Figure 1F and G)**.

We thank the reviewer for suggesting to sort the different cell populations in the mosaic patient. We FACS sorted neutrophils based on their capacity to produce reactive oxygen as indicated by the DHR assay following PMA stimulation. To maintain viable cells after the sort, we performed a submaximal stimulation that still allowed clear separation of the ~70% DHR low/negative cells and the ~30% DHR positive cells **(Figure 1H)**. We now show that neutrophils from the patients with normal DHR+ profile show 100% wild type variants whereas DHR-low/negative neutrophils show 98% pathogenic variants **(Figure 1I)**. To further confirm that the DHR+ and DHR-low/negative neutrophils in this patient have different functional characteristics, we performed functional characterisation of the FACS sorted cells. Given that reactive oxygen species are used by phagocytes to fight phagocytosed bacteria, we functionally tested the ability of DHR+ and DHR- neutrophils of the patient to kill bacteria in a gentamicin protection assay. We found that DHR+ neutrophils from the patient had a similar bacterial handling capacity as the control cells (quantified as colony forming units), while the DHR- cells were less able to eliminate bacteria **(Figure 1J)**.

In summary, we have shown that the FACS sorted DHR+ and DHR- populations are functionally distinct in terms of bacterial handling and that the mosaicism arose early on in embryonic development so that the mutation is present in both mesoderm- and ectoderm-derived cells.

Figure 1

Reviewer #3 (Remarks to the Author):

Gonçalves Serra and colleagues investigate genetic factors contributing to inflammatory bowel disease (IBD) in a cohort of 145 very-early-onset (VEO) patients (onset <6 years) in whom no Mendelian disorders were clinically suspected. Firstly, they screen for rare, disruptive mutations in 67 genes known to be associated with Mendelian disorders with IBD-like inflammatory phenotypes ascertained from the literature. They identify five patients from four families with mutations in the genes CYBB, CYBA, XIAP and SH2D1A. Three mutations are on the X-chromosome in males of which one is a mosaic de novo mutation affecting 70% of cells in all immune cell subsets. Detailed functional evidence is provided for these mutations. Subsequently, authors calculate a polygenic risk score for known genetic risk factors for CD and UC and demonstrate that the polygenic risk score for CD and UC is significantly increased in the VEO cases compared to healthy controls. They conclude that the polygenic component underlying adult IBD contributes to the VEO cases as well, which is in line with the identification in previous GWAS of risk loci shared between pediatric and adult-onset IBD.

The manuscript is elegantly written and reads very well. Authors took great care in the genetic and functional evaluation of candidate mutations, and evidence for a pathogenic status of the reported variants is strong. Results broaden the range of clinical presentations and phenotypes associated with mutations in these genes. This is of high importance for the clinical care of VEO-IBD patients. Somatic mosaicism for CYBB has been demonstrated previously for the immunodeficiency CGD but the current report adds to the growing support of somatic mutations in autoimmune disease. Limitations of the WES part to a subset of genes and of the design of the polygenic risk score part limit to some extent novelty towards the wider scientific community.

Major comments:

1) WES is performed, but analysis is limited to 67 genes already known to be associated with Mendelian

disorders with IBD-like inflammatory phenotypes ascertained from the literature. Has evaluation of other genes (eg. all PID genes) been possible, will they be reported elsewhere and/or undertaken as a future perspective?

We have performed several strategies to identify additional IBD associated genes and/or pathways.

One strategy was to search for homozygous/potential compound heterozygous or hemizygous essential loss-of-function variants in patients. This screen revealed a homozygous stop variant in *PCSK1* (p.R391X). *PCSK1* (Proprotein Convertase Subtilisin/Kexin type 1) encodes the proprotein convertase enzyme which cleave prohormones. Clinical syndromic features of the patient included IBD-unclassified before the age of 1. Interestingly, the initial intestinal inflammation did not progress, but the phenotype did change over time. After patient recruitment and submission of the DNA sample for sequencing, the phenotype evolved towards growth delay, excessive weight gain, and endocrine disorders including diabetes and hypothyroidism, hypogonadism, cryptorchism, cortisol deficiency and chronic lung disease. Whereas the initial phenotype was uncharacteristic, the subsequent syndromal findings are fully explained by *PCSK1* deficiency. This finding highlights the value of next generation sequencing as a predictive diagnostic tool as well as the need to take phenotype progression into account. No other likely essential loss-of-function gene variants were identified. This result has been added to the main text.

In our original submission we did conduct exome-wide gene-based association tests to identify new VEO-IBD risk genes with an increased burden of rare variants in cases versus a large set of controls (N=3,588). However, this strategy did not reveal new VEO-IBD genes and thus chose to present this work in the Supplementary material to give more space to our main significant findings. Following this comment by the reviewer we have now moved it to the main text and expanded it to better highlight these analyses. Briefly, nine different exome-wide gene-based screens were conducted using different variant inclusion criteria (for variant severity and minor allele frequency). No individual gene achieved exome-wide significance irrespective of the variant inclusion criteria.

We next conducted rare-variant burden tests across multiple related genes, such as PID genes, or those that reside in the same biological pathway. This approach offers additional statistical power compared to individual gene tests, as a larger number of variants are collapsed across a larger testing unit. In total we tested 195 different biological genesets, 186 of which represent the whole set of KEGG pathways available in the KEGG pathway database. No geneset or pathway showed a significant burden of rare variants in patients versus controls after correction for multiple testing.

Obviously we cannot exclude additional pathogenic variants but like to stress that those would require extensive functional validation and replication by additional patients in other cohorts.

2) In the discussion of disease-causing variants in pediatric IBD, some very recent advance-online publications are missing (e.g. Gastroenterology issues of Feb and March 2018).

We thank the reviewer for pointing these out and we added the publications (Amininejad L, et al. *Gastroenterology*. 2018 Jun;154(8):2165-2177. and Denson LA, et al. *Gastroenterology*. 2018 Jun;154(8):2097-2110) to the main text.

3) The polygenic risk score for CD and UC is highly significantly increased in VEO-IBD compared to healthy controls. Conclusions that can be drawn from the comparison of VEO versus adult-onset cases are limited as the VEO cases are not subdivided in CD or UC like the adult-onset cases and by the sample size of 99 patients from this single cohort. Similarly, these factors limit the correlation with age at onset.

We thank the reviewer for suggesting these additional analyses. We have now conducted the suggested additional analysis restricting the CD polygenic risk score (PRS) tests to those VEO-IBD cases defined as CD or CD+IBDu (indeterminate IBD), and the UC PRS tests restricted to those with UC or UC+IBDu (indeterminate IBD). The results

from these analyses replicate the finding of our original analysis - case groups have significantly higher polygenic burdens than controls. We have expanded the results section of the main text to include these additional analyses.

4) In the discussion, authors hypothesize that the extent to which the polygenic background influences VEO-IBD susceptibility varies depending on the monogenic foreground. Their data do not allow addressing this hypothesis, as they indeed indicate.

We thank the review for pointing this out and note that reviewer 1 raised a similar concern. In the revised manuscript we have edited the wording to make it more clear that we are not claiming that "*the extent to which the polygenic background influences VEO-IBD susceptibility varies depending on the monogenic foreground*" is a finding of our paper. Instead, we are discussing potential models for the genetic architecture of VEO-IBD, how they would compare to other rare and early-onset disease, and importantly, the experiments that would be needed to evaluate these. As rightly pointed out above, comparing the PRS of VEO-IBD patients with and without monogenic variants constitutes one of these experiments.

Minor comments:

1) Description of patient cohort: how many patients were excluded from the original cohort because of a clinical diagnosis of a known Mendelian disease was suspected and confirmed? Together with the currently found mutations this would provide an estimate of the Mendelian contribution to VEO-IBD at least in this particular dataset.

The total number of patients with Mendelian disorders that have been diagnosed previously because of clinical suspicion can only be estimated. As previously reported, at least 19 cases were identified at one referral center in a cohort of 62 patients presenting before the age of 2 years (Kammermeier et al. JCC 2017). Since this was not an epidemiological study, we have not performed a systematic retrospective analysis of the proportion of patients that were diagnosed previously with a Mendelian disorder (bone marrow transplanted, treated with conventional therapy or deceased).

2) Ancestry of the mutation carriers should be discussed in the Results section as several appear of non-European ancestry.

In the revised manuscript we now describe the ancestries of the patients carrying clinically relevant VEO-IBD mutations in the main text.

3) A conservative INFO score threshold of 0.9 is set for the polygenic risk score. What is the proportion of missing data resulting from the application of this threshold and how were missing data handled? Is there a difference in missing rate between VEO cases and adult cases and controls as they were obtained from different sources?

The INFO threshold of 0.9 used in the PRS analysis is indeed conservative. This threshold was selected given the small size of the VEO-IBD cohort. The proportion of missing data, overall across all the backbone SNPs used in the PRS analysis, was investigated and was not significantly different across VEO-IBD cases, adult cases and controls (after removing samples that had outlier missingness rates). This point has now been added to the revised manuscript.

The initial backbone of SNPs that were available for the PRS analysis (i.e. those that showed at least suggestive evidence of association ($P < 1 \times 10^{-5}$) with CD or UC in the European cohort of Liu et al 2015) was 174 and 146 for CD and UC, respectively. Of these, only 42% (76/181) and 46% (71/154), respectively, were present in the genotype data post INFO filtering. We then selected proxy markers with an $r^2 > 0.9$ to substitute missing SNPs, which led to a

total of 147 CD and 119 UC-associated alleles available in both the VEO-IBD and the UKIBDGC genotype data. A total of 71 and 48 SNPs for CD and UC, respectively, were substituted with proxy SNPs.

4) Some references are incomplete.

We thank the reviewer for pointing this out and apologise for this. We have checked that the bibliography list is now correctly formatted.

Reviewers' Comments:

Reviewer #1:

Remarks to the Author:

While the authors have addressed one of my comment with the new data (presented as Figure 1H-J), the main problem still remains, which refers to the lack of either novelty or rigor of this study. The 3 findings of this study are: 1) a somatic mosaicism in CYBB gene of a patient with VEO-IBD; 2) five patients with VEO-IBD carrying mutations in 4 genes that are previously very well known to play a role in monogenic-IBD; 3) operation of a polygenic component in VEO-IBD.

My three main concerns regarding these findings, and this study overall are:

Lack of rigor: Authors have found a case of somatic mosaicism in CYBB, where 70% of the buccal and blood cell subsets carried a hemizygous nonsense allele while the rest of the cells showed normal genotype at the locus, which ultimately lead them to conclude somatic mosaicism as a non-classical mode of inheritance of VEO-IBD.

However, this was found in 1 patient with VEO-IBD. In fact a similar case of somatic mosaicism in CYBB has previously been reported in 2 unrelated patients with chronic granulomatous disease (CGD); Yamada et al., *Gene*; 2012). As I stated in my initial review, this finding is extremely preliminary at this stage. In order to facilitate insights into the benefit of profiling somatic variation in diseases like IBD, a well-designed, comprehensive study is needed. The current study by Gonçalves Serra et al. would certainly not fit this criteria. This patient could in fact be a case of CGD, but, may have received a misdiagnosis. Looking at just the history of invasive infections to rule out the possibility of CGD might not be a comprehensive approach to claim that this patient has no CGD.

Lack of novelty: Again, as I pointed in my initial review, the finding of 5 VEO-IBD patients with mutations in 4 genes that are very well known to play a role in monogenic IBD is essentially a confirmation of what's already been known rather than a novel discovery. In response to my comment, authors have stated that – even though we did not identify any novel VEO-IBD-genes, our findings “broaden the range of clinical presentations and phenotypes associated with mutations in these genes”. To be honest, I don't completely understand what this means, when their findings are essentially a replication of what's already been known.

Issues with small sample size and the lack of validation of the finding of the operation of a polygenic component in VEO-IBD: Authors have computed genetic risk scores based on the previously known common variants of late-onset CD and UC to demonstrate that common genetic variation associated with adult onset IBD also plays a role in VEO-IBD. However, this conclusion is limited by the total sample size of 99 patients from this cohort. These numbers get even smaller when the VEO-IBD patients are subdivided into VEO-CD and VEO-UC.

This raises the question that if there are no underlying monogenic causes, and considering that common variants of little effect sizes contribute to VEO-IBD – how do the authors explain why all these patients have a severe disease? This to me sounds more of an issue of defects in the immune system in general, rather than a role for common variants as concluded by the authors. For instance, both B- and T-cell development, migration, and proliferation can drive disease in VEO-IBD. Lack of immune work-up on this cohort is another major limitation.

Reviewer #3:

Remarks to the Author:

The additional data and additions to the text sufficiently address my previous questions and

comments and substantially clarify the message further. No further comments.

Reviewer #4:

Remarks to the Author:

Serra et al present NGS data for 145 VEO-IBD patients and identify 6 potential causative mutations. They identify 1 case of a potential somatic mosaicism and examine a polygenic risk score. The sequencing effort is not much different than 125 VEO-IBD patients by Kelsen et al 2015 (Gastro), the functional studies show functional defects that have been previously described and the genetics risk score is underpowered and not validated in another VEO-IBD cohort.

Issues:

The general organization and Aims of the paper were confusing as written. Would expect the first paragraph of the results to describe the cohort and type of NGS used for this study, as well as data related to sequencing was carried out in trios. Although some of the information is found in the supplemental material, it is very difficult to understand what the goal of this paper is. Perhaps the most interesting aspect of the paper was the utility of WES (4%) to identify monogenic forms of VEOIBD was completely ignored.

Without any description of the study, the authors immediately describe a case of a 31-year old male? Was this a VEO-IBD patient? As previously stated in review, these findings are not novel for CGD. Also, the following statement is already known "This is an important insight into the potential utility of gene therapy for treatment of CGD; correcting the genetic sequence in around 30% of phagocytes could be sufficient to prevent serial life threatening infections, but likely insufficient to treat VEO-IBD" is already known. The authors should cite JCI 2018 paper regarding p40phox.

Functional studies are generally convincing to demonstrate the defect.

Gene-based association analysis: it is unclear if all the patients were used for this and unclear why this section is added as does not add much information. Would not expect any association with small number of patients and diverse ethnicity and phenotype. Are these compared to Caucasian samples? Why not compare WES data with other cohorts if gene risk discovery is an aim. Same comment for the biological pathways as there this is a small diverse sample size.

Polygenic risk score: it is unclear the utility of this approach. How were the SNPs identified: WES, the genotyping, imputation? It is unclear if the non-Caucasian individual were removed or accounted for in this analysis as they could explain the observed differences. The authors should attempt to validate the PRS in another cohort of VEOIBD patients as this should be feasible within the IIBDGC.

Overall the authors themselves admit they are under powered to detect the genetic variables studied here and lack the suggested novelty of the manuscript. Therefore, these results should be considered as preliminary. The authors may discuss the utility of trio analysis to improve diagnosis.

Reviewer #5:

Remarks to the Author:

This is a well written manuscript that addresses a clinically important issue. A Cohort of 145 patients with patients with very only onset (VEO) inflammatory bowel disease (IBD), who have no suspected Mendelian etiology, is investigated. Already by a candidate gene analysis (67 candidate genes) they identify 5 patients with variants in CVBA, CYBB, XIAP and SH2D1A. Interestingly one

variant represents a new mutation that in a mosaic affects 70% of immune cells in various subsets. They also use the cohort to show that genetic overall risk contributes to VEO IBD in a similar way to adult onset IBD.

With no doubt this manuscript represents a solid piece of translational medicine. Although it does not provide the novelty of new discovery or a principal and in-depth exploration of a biologic system (immunodeficiency due to mosaic mutation in CYBB have been described before, the candidate gene approach is somewhat limited and the risk score applied is not novel) it makes a widely visible contribution to clinical advancement in the field.

Reviewers' comments:

Reviewer #1 (Remarks to the Author):

While the authors have addressed one of my comment with the new data (presented as Figure 1H-J), the main problem still remains, which refers to the lack of either novelty or rigor of this study. The 3 findings of this study are: 1) a somatic mosaicism in *CYBB* gene of a patient with VEO-IBD; 2) five patients with VEO-IBD carrying mutations in 4 genes that are previously very well known to play a role in monogenic-IBD; 3) operation of a polygenic component in VEO-IBD.

My three main concerns regarding these findings, and this study overall are:

Lack of rigor: Authors have found a case of somatic mosaicism in *CYBB*, where 70% of the buccal and blood cell subsets carried a hemizygous nonsense allele while the rest of the cells showed normal genotype at the locus, which ultimately lead them to conclude somatic mosaicism as a non-classical mode of inheritance of VEO-IBD.

However, this was found in 1 patient with VEO-IBD. In fact a similar case of somatic mosaicism in *CYBB* has previously been reported in 2 unrelated patients with chronic granulomatous disease (CGD); Yamada et al., *Gene*; 2012). As I stated in my initial review, this finding is extremely preliminary at this stage. In order to facilitate insights into the benefit of profiling somatic variation in diseases like IBD, a well-designed, comprehensive study is needed. The current study by Gonçalves Serra et al. would certainly not fit this criteria. This patient could in fact be a case of CGD, but, may have received a misdiagnosis. Looking at just the history of invasive infections to rule out the possibility of CGD might not be a comprehensive approach to claim that this patient has no CGD.

The reviewer correctly points out that that somatic mosaicism in *CYBB* has been previously reported in two patients with clinical presentation of chronic granulomatous disease.

There are two main differences between our result and those from Yamada et al, *Gene*, 2012, the most important of which is the sequence of discovery. We identified the “heterozygous genotype” in the X-linked gene in a patient without clinical signs of CGD, and validated this by next-generation sequencing as well as DHR functional assays. To our knowledge this is the first identification of a mosaicism that has been identified by sequencing in a patient with IBD. The previous study of Yamada et al, *Gene*, 2012 performed functional tests that were warranted by clinical signs of CGD. Our identification suggests that more could and should be done to identify mosaic causal events in VEO-IBD patients.

A major difference between our result and that of Yamada et al, is the point at which the somatic mutation arose during embryogenesis. Yamada et al describe two patients with CGD who have mosaicism in *CYBB*, and grow out clones to identify the percentage of wild-type *CYBB* cells. In their first patient, this percentage varies from 0% in PBMCs (derived from mesoderm) to 1.8% in

the buccal swab (predominantly ectoderm but with a small percentage of mesoderm-derived immune cells). For their second patient, the percentage of wild-type CYBB cells varies from 1.6% in PBMCs to 18.8% in the buccal swab. In our study, the proportion of cells carrying wild-type CYBB is much higher, ranging between 24-37% in all surveyed immune cell types (neutrophils, monocytes, CD4+ T cells, CD20+ B cells and NK cells) and 10-65% of cells from buccal swabs (mean 41.5%). We have also included hair follicle cells in order to rule out the contribution of immune cells present in the buccal swab (36% wild type). This suggests that the mutation in our patient occurred earlier in embryogenesis than those reported by Yamada et al., but still prior to the separation of the mesoderm and ectoderm layers at day 9. The mosaicism we observe is unlikely to be driven by clonal expansion because CYBB is not expressed in epithelia and thus cells carrying this mutated allele would be unlikely to have the same selective advantage as immune cells. Furthermore, a reversion event is unlikely in our patient since the mutated variant is not carried by the mother of our patient (the probability that there were both a de-novo mutation causing the pathogenic variant in oocytes, or mosaicism in the mother affecting only her reproductive organs, followed by a reversion event in the patient early in embryogenesis is vanishingly small).

Overall, this patient does not have the pathognomonic immunodeficiency features characteristic of chronic granulomatous disease (absent infections) but has intestinal inflammation. This is part of a genetic and phenotypic disease spectrum.

Lack of novelty: Again, as I pointed in my initial review, the finding of 5 VEO-IBD patients with mutations in 4 genes that are very well known to play a role in monogenic IBD is essentially a confirmation of what's already been known rather than a novel discovery. In response to my comment, authors have stated that – even though we did not identify any novel VEO-IBD-genes, our findings “broaden the range of clinical presentations and phenotypes associated with mutations in these genes”. To be honest, I don't completely understand what this means, when their findings are essentially a replication of what's already been known.

We certainly agree with the reviewer that the identification of five VEO-IBD patients carrying mutations in known VEO-IBD genes is not a novel scientific discovery. However, we do believe our manuscript describes several important novel elements and discoveries. These include:

1. The first study to investigate a cohort of patients with very early onset IBD not only by next generation sequencing (genome/exome/targeted-panel sequencing) but also by genotyping of common variants.
2. The first identification of somatic mosaicism causal for VEO-IBD via exome-sequencing, an important discovery for how such studies should be analysed going forwards (i.e. investigators should explicitly search for mosaic events within known VEO-IBD to increase diagnostic yield). This is an important consideration at the earliest stages of genotype calling and quality controlling exome sequencing data.
3. The first to establish a polygenic component operative in VEO-IBD, and we have now replicated this discovery in an independent VEO-IBD cohort and independent control set.

This analysis has major implications for our understanding of the genetic architecture of VEO-IBD. For example, it is clear that whatever is causing these children to have very early onset and severe disease is doing so on a genetic background predisposed to IBD. If we had observed that VEO-IBD cases had an IBD PRS similar to our population controls, this would have suggested that the disease is likely to be caused by rare genetic or environmental effects and that lessons learned from adult onset IBD would be irrelevant for understanding and treating VEO-IBD. This discovery also aids the design of future genetic association studies into age-at-onset of IBD, because there is now justification for including VEO-IBD patients alongside other age groups in these studies.

4. Despite being well-powered to do so, we do not replicate the previously reported association between VEO-IBD and carriage of rare variants in known primary immune deficiency genes (Kelsen et al 2015).

To clarify the extensive genetic analyses described in this paper (which is beyond any previously published VEO-IBD cohort) we now provide a summary figure (included in Supplementary Material, Supplementary Figure 1):

One question raised by the reviewer is whether the mutation in CYBB just reveals a misdiagnosis of CGD. We certainly agree that there is a spectral overlap between loss of function (LOF) variants in the NADPH oxidase complex causing immunodeficiency and IBD-like inflammation on the one hand and hypomorphic NADPH gene variants seen in IBD patients without immunodeficiency (Denson LA et al Gastroenterology 2018; Dhillon SS et al Gastroenterology 2014). Our finding of mosaicism now adds another and novel dimension to

this spectrum by showing that the proportion of cells with complete LOF in CYBB also likely affects the phenotype spectrum, i.e. 30% wild type neutrophils are sufficient to prevent infection but 66% of macrophages that cannot kill bacteria and produce inflammatory cytokines are a key risk factor for intestinal inflammation.

Issues with small sample size and the lack of validation of the finding of the operation of a polygenic component in VEO-IBD: Authors have computed genetic risk scores based on the previously known common variants of late-onset CD and UC to demonstrate that common genetic variation associated with adult onset IBD also plays a role in VEO-IBD. However, this conclusion is limited by the total sample size of 99 patients from this cohort. These numbers get even smaller when the VEO-IBD patients are subdivided into VEO-CD and VEO-UC.

We disagree with the reviewer that our study has issues (of low statistical power) due to small sample size. In our case, we used the T-test to determine if there is a statistically significant difference in the mean PRS in VEO-IBD cases versus population controls. The T-test p-values from these analyses, considering common risk variants associated with either CD or UC, are $\sim 4 \times 10^{-10}$ and $\sim 1 \times 10^{-10}$, respectively. Both of these results far exceeds even the most conservative significance threshold, and thus it is hard to claim that our conclusion that a polygenic component contributes to VEO-IBD “is limited by the total sample size”. We strongly believe that this is a robust, novel and important discovery for our understanding of how genetic variation contributes to VEO-IBD risk.

We agree with the reviewer that validation is key and have therefore performed a validation experiment with independent VEO-IBD cases and controls. We used a cohort of 117 VEO-IBD patients from the SickKids Toronto IBD cohort and 2603 controls from the NIDDK Genetics Consortium (all of European-descent). Quality control, genome-wide SNP imputation and the construction of the polygenic risk scores was performed using the same pipeline applied to our COLORS-in-IBD cohort. We are pleased to report that we also observed elevated IBD PRS' in the SickKids Toronto VEO-IBD cohort compared to NIDDK controls ($P = 4.60 \times 10^{-10}$ and $P = 4.32 \times 10^{-12}$ for CD and UC, respectively). We have changed Figure 3 in the manuscript to include both the discovery and the replication findings and have added a description of this cohort, and our analysis of it, to the Methods and Results sections.

This raises the question that if there are no underlying monogenic causes, and considering that common variants of little effect sizes contribute to VEO-IBD – how do the authors explain why all these patients have a severe disease? This to me sounds more of an issue of defects in the immune system in general, rather than a role for common variants as concluded by the authors. For instance, both B- and T-cell development, migration, and proliferation can drive disease in VEO-IBD. Lack of immune work-up on this cohort is another major limitation.

We agree with the reviewer that the cause of VEO-IBD in the majority of the patients in our cohort remains unknown and that this is an outstanding research question. We are reluctant to speculate on the biological mechanisms that may be driving this but, as we conclude in our manuscript, whatever is causing the severe early-onset disease in these children (be it genetic or otherwise) is doing so on a genetic background that is predisposed to IBD. To be clear, we are certainly not concluding that the VEO-IBD in these patients is *caused* by common genetic variants, but our data suggest that these variants do contribute to the presentation of the

disease. We agree with the reviewer that B- and T-cell development, migration, and proliferation could drive disease in VEO-IBD but we believe these cellular traits could, in part, be influenced by common genetic variants (and/or rare genetic variants).

It is outside the scope of our current genetic study to undertake a detailed immune work-up in the cohort. Importantly, we have undertaken functional work-up for the five VEO-IBD patients for who we do discover pathogenic variants.

Reviewer #3 (Remarks to the Author):

The additional data and additions to the text sufficiently address my previous questions and comments and substantially clarify the message further. No further comments.

We are delighted that the reviewer approves of the changes we made following the initial peer reviews and agrees with us that our message is clear.

Reviewer #4 (Remarks to the Author):

Serra et al present NGS data for 145 VEO-IBD patients and identify 6 potential causative mutations. They identify 1 case of a potential somatic mosaicism and examine a polygenic risk score. The sequencing effort is not much different than 125 VEO-IBD patients by Kelsen et al 2015 (Gastro), the functional studies show functional defects that have been previously described and the genetics risk score is underpowered and not validated in another VEO-IBD cohort.

Issues:

The general organization and Aims of the paper were confusing as written. Would expect the first paragraph of the results to describe the cohort and type of NGS used for this study, as well as data related to sequencing was carried out in trios. Although some of the information is found in the supplemental material, it is very difficult to understand what the goal of this paper is. Perhaps the most interesting aspect of the paper was the utility of WES (4%) to identify monogenic forms of VEO-IBD was completely ignored.

We describe the cohort, and the genotyping, in detail in the methods section of the manuscript. We only whole-exome sequenced the affected VEO-IBD patients and we did not perform sequencing of trios. However, we genotyped variants of interest in some parents and siblings to confirm genetic transmissions.

To clarify the manuscript, we have added some results of our WES data generation to the beginning of the results section. Importantly, we have now provided a supplemental figure (Supplementary Figure 1) that summarises our study protocol (various analyses and QC conducted). If the editor believes that a more substantial change is required for the manuscript to be accepted for publication, we would happily follow their guidelines.

Without any description of the study, the authors immediately describe a case of a 31-year old male? Was this a VEO-IBD patient? As previously stated in review, these findings are not novel for CGD. Also, the following statement is already known “This is an important insight into the potential utility of gene therapy for treatment of CGD; correcting the genetic sequence in around 30% of phagocytes could be sufficient to prevent serial life threatening infections, but likely insufficient to treat VEO-IBD” is already known. The authors should cite JCI 2018 paper regarding p40phox.

We thank the reviewer for these comments. Indeed, we stated in the patient description that the patient had infantile onset IBD: “The initial screening for pathogenic variants in established monogenic IBD genes identified a nonsense mutation in *CYBB* (p.W380X) in a 31 year old male patient of European descent with *infantile-onset* of granulomatous colitis, perianal abscesses and hidradenitis suppurativa.”

We have explained in the comments to reviewer 1 (please see above) that there are significant differences compared to the previously published CGD patients with mosaicism. Firstly, identification of patients with *CYBB* mosaicism via sequencing has never been shown before. This has immense therapeutic implications since it indicates that it is possible that other VEO-IBD and IBD cohorts have mutations in genes associated with monogenic disorders, but that this could have been missed. In addition, we clearly determine the timing of the somatic mutation during embryogenesis. Whereas Yamada et al speculate about two different mechanisms to lead to a variable mosaicism, in our patient there is a median of a third of cells that are wild type from different embryonic lineages. Furthermore, we know that this mutation was not carried by the patient’s mother. Taken together, we can thus conclude that this is very likely a de novo mutation that arose early in embryogenesis, likely between days two and five, and certainly before day nine, the time at which the mesoderm and ectoderm separate.

We thank the reviewer for pointing us to the important van der Geer et al. JCI 2018 paper. While the symptoms in patients with *NCF4* mutations were milder than classical CGD, the symptoms still indicated CGD due to the presence of peripheral infections. Interestingly, in those patients the DHR assay was not informative (which is different to our findings in the mosaicism patient). The JCI paper adds to the aforementioned evidence that there is large phenotypic spectrum of NADPH oxidase deficiencies depending on the defective gene and extent of somatic mosaicism. Moreover, the reported *NCF4* mutations are loss-of-function (with one hypomorphic exception). Our patient presents with mosaicism in *CYBB*, in which mutations usually cause severe disease. These two cases are entirely different genetically, one presenting a milder form of disease due to a different protein being affected, whereas our patient represents a rare case of somatic mosaicism with no symptoms of disease. It is certainly not already known that having 30% functioning neutrophils can be sufficient to protect from CGD - this is a novel finding of our work.

Functional studies are generally convincing to demonstrate the defect.

We are pleased to read that the reviewer is happy with our functional studies.

Gene-based association analysis: it is unclear if all the patients were used for this and unclear why this section is added as does not add much information. Would not expect any association with small number of patients and diverse ethnicity and phenotype. Are these compared to Caucasian samples? Why not compare WES data with other cohorts if gene risk discovery is an aim. Same comment for the biological pathways as there this is a small diverse sample size.

Polygenic risk score: it is unclear the utility of this approach. How were the SNPs identified: WES, the genotyping, imputation? It is unclear if the non-Caucasian individual were removed or accounted for in this analysis as they could explain the observed differences. The authors should attempt to validate the PRS in another cohort of VEOIBD patients as this should be feasible within the IIBDGC.

The answers to these questions were supplied in the Methods section of manuscript:

1. We assessed the contribution of IBD loci associated with adult-onset IBD to VEO-IBD. This analysis made use of the genome-wide genotype and imputation data, rather than our exome-sequences, because the vast majority of IBD associations identified via GWAS are located in intergenic or intragenic regions and are best tagged by non-coding variants.
2. The SNP set for the risk score calculation was generated by selecting the most associated SNP showing at least suggestive evidence of association ($P < 1 \times 10^{-5}$) with CD or UC in the European cohort of the largest IBD GWAS meta-analysis at the time of the analysis. This resulted in 174 and 146 alleles associated with CD or UC, respectively. Proxy markers with an $r^2 > 0.9$ were selected to substitute SNPs that were absent from the VEO-IBD and/or UKIBDGC genotype data. Post QC, a total of 147 CD and 119 UC-associated alleles were available in both the VEO-IBD and the UKIBDGC genotype data.
3. To minimize population stratification we restricted our risk score analysis to European VEO-IBD cases of high genotype quality (N=99 COLORS in IBD cohort and N=117 SickKids cohort); the same was done for control datasets used in these comparative analyses.

To increase clarity, we have add some details of the methods to the Results section of the manuscript.

As suggested by the reviewer, we have successfully validated the VEO-IBD PRS score in another cohort (SickKids Toronto VEO-IBD cohort) and using an independent cohort of controls. For more details, please see our response to the third comment made by the first reviewer. As an aside, the IIBDGC has genotype data on very few infantile-onset cases.

Overall the authors themselves admit they are under-powered to detect the genetic variables studied here and lack the suggested novelty of the manuscript. Therefore, these results should be considered as preliminary. The authors may discuss the utility of trio analysis to improve diagnosis.

Throughout our manuscript we were careful to interpret our results in light of our statistical power. However, we disagree with the reviewer that we are underpowered to “detect the genetic variables studied here”. As stated in response to the third comment of Reviewer 1, we are extremely well-powered to detect the increased burden of common genetic risk variants seen in VEO-IBD cases versus population controls (this is evidenced by the fact our P-values were $\sim 4 \times 10^{-10}$ and $\sim 1 \times 10^{-10}$, for CD and UC, respectively). We also replicated our discovery of a polygenic component operative in VEO-IBD using an independent cohort of cases and controls. Thus, we do not believe our results should be considered preliminary.

Reviewer #5 (Remarks to the Author):

This is a well written manuscript that addresses a clinically important issue. A Cohort of 145 patients with patients with very only onset (VEO) inflammatory bowel disease (IBD), who have no suspected Mendelian etiology, is investigated. Already by a candidate gene analysis (67 candidate genes(they identify 5 patients with variants in CVBA, CYBB, XIAP and SH2D1A. Interestingly one variant represents a new mutation that in a mosaic affects 70% of immune cells in various subsets. They also use the cohort to show that genetic overall risk contributes to VEO IBD in a similar way to adult onset IBD.

With no doubt this manuscript represents a solid piece of translational medicine.

Although it does not provide the novelty of new discovery or a principal and in-depth exploration of a biologic system (immunodeficiency due to mosaic mutation in CYBB have been described before, the candidate gene approach is somewhat limited and the risk score applied is not novel) it makes a widely visible contribution to clinical advancement in the field.

We would like to thank the reviewer for acknowledging the important contribution that our paper makes to the clinical advancement of VEO-IBD management.